# Spatial transcriptomics in the adult *Drosophila* brain and body

**Jasper Janssens[1,2], Pierre Mangeol[3†], Nikolai Hecker[1,2,4†], Gabriele Partel[1,2,4†], Katina I Spanier[1,2,4], Joy N Ismail[1,2], Gert J Hulselmans[1,2,4], Stein Aerts[1,2,4*‡], Frank Schnorrer[3*‡]**

[1]VIB-KU Leuven Center for Brain and Disease Research, KU Leuven, Leuven, Belgium; [2]Laboratory of Computational Biology, Department of Human Genetics, KU Leuven, Leuven, Belgium; [3]Aix Marseille University, CNRS, IBDM, Turing Centre for Living Systems, Marseille, France; [4]VIB Center for AI & Computational Biology, KU Leuven, Leuven, Belgium

**\*For correspondence:**
stein.aerts@kuleuven.be (SA);
frank.schnorrer@univ-amu.fr (FS)

[†]These authors contributed equally to this work
[‡]These authors also contributed equally to this work

**Competing interest:** The authors declare that no competing interests exist.

## eLife Assessment

This **important** study presents a method to visualize the location of the cell types discovered through single-cell RNA sequencing. The data allowed the authors to build spatial tissue atlases of the fly head and body, and to identify the location of previously unknown cell types. The data are **convincing** and appropriate, and the authors validate the methodology in line with the current state-of-the-art.

**Abstract** Recently, we have achieved a significant milestone with the creation of the Fly Cell Atlas. This single-nuclei atlas encompasses the entire fly, covering the entire head and body, in addition to all major organs. This atlas catalogs many hundreds of cell types, of which we annotated 250. Thus, a large number of clusters remain to be fully characterized, in particular in the brain. Furthermore, by applying single-nuclei sequencing, all information about the spatial location of the cells in the body and of about possible subcellular localization of the mRNAs within these cells is lost. Spatial transcriptomics promises to tackle these issues. In a proof-of-concept study, we have here applied spatial transcriptomics using a selected gene panel to pinpoint the locations of 150 mRNA species in the adult fly. This enabled us to map unknown clusters identified in the Fly Cell Atlas to their spatial locations in the fly brain. Additionally, spatial transcriptomics discovered interesting principles of mRNA localization and transcriptional diversity within the large and crowded muscle cells that may spark future mechanistic investigations. Furthermore, we present a set of computational tools that will allow for easier integration of spatial transcriptomics and single-cell datasets.

## Introduction

Single-cell technologies have revolutionized biological research, allowing researchers for the first time to study complex tissues with unprecedented resolution (*Klein et al., 2015*), leading to the creation of tissue-specific atlases (*Schaum et al., 2018*; *The Tabula Sapiens Consortium, 2022*). However, upon tissue dissociation into single cells, the spatial context of the cell in the organism and of the mRNAs within the cell is lost, since all data of the cell are compressed into one multidimensional data point. This is a significant drawback, as the localization of a cell type within a tissue can inform about the function of this cell (*Tomancak et al., 2002*). Furthermore, many mRNAs show subcellular localization patterns that may have functional roles (*Jambor et al., 2015*; *Lécuyer et al., 2007*).

This loss of information can be circumvented using spatially resolved transcriptomics (SRT). In SRT, spatial information of the RNA species is determined either through the use of spatially localized barcoded DNA arrays with or without the use of microfluidics, followed by sequencing (*Rodriques et al., 2019*; *Wang et al., 2022*), or by imaging using multiplexed rounds of single-molecule fluorescence in situ hybridization (smFISH) methods. While barcoded sequencing approaches allow unbiased mapping of all transcripts, their spatial resolution is, in general, lower (*Moses and Pachter, 2022*). Multiplexed smFISH identifies the position of each RNA molecule independently at a high spatial resolution, but it is limited in throughput to a few genes. Recently, technical breakthroughs have been presented to increase the number of genes to several hundred through the use of barcoding (*Chen et al., 2015*; *Eng et al., 2019*; *Lubeck et al., 2014*; *Zhang et al., 2021*).

The adult fruit fly *Drosophila melanogaster* is a widely used model for cell biology, neurobiology, physiology and behavior (*Sokolowski, 2001*; *Grenier and Leulier, 2020*; *Benton, 2022*). The recent completion of the entire adult fly cell atlas (FCA) (*Li et al., 2022*), and the aging fly cell atlas (*Lu et al., 2023*) provided an in-depth annotation of various cell types, however without information about the localization of these cells or their mRNAs. In addition, many single-cell clusters were left unannotated although marker genes were identified.

One tissue that would benefit greatly from SRT are the large polynucleated muscle cells. In tubular head and leg muscles, the nuclei are located in a central row surrounded by contractile myofibrils, while in fibrillar indirect flight muscles (IFMs), the nuclei are organized in multiple rows between the myofibril bundles (*Luis and Schnorrer, 2021*; *Schönbauer et al., 2011*), distributed along the length of the large 1 mm long fibers (*Spletter et al., 2018*). This organization is thought to optimize the mRNA distribution by minimizing transport distances (*Bruusgaard et al., 2003*) and misalignment of the nuclei has been linked to severe myopathies in humans (*Folker and Baylies, 2013*). While single nuclear (nc)RNA-seq allows studying of potentially different types of nuclei in the muscle, earlier studies have proposed classical FISH to study and describe mRNA localization in muscle fibers (*Denes et al., 2021*; *Pinheiro et al., 2021*).

Here, we applied Molecular Cartography (MC, Resolve Biosciences), a multiplexed single-molecule hybridization method with submicron resolution (~136 nm), to create a spatial map of gene expression of the entire head and body of adult *Drosophila melanogaster* flies. While embryonic development had already been studied using SRT (*Wang et al., 2022*), we present the first high-throughput smFISH-based SRT dataset of the various tissues of the adult fly. We compare the data with existing RNA atlases, using our spatial data to validate annotations in the body and identify the location of uncharacterized cell clusters in the brain. Furthermore, we reveal subcellular mRNA localization patterns in the flight muscle cells and regionalized RNA expression in the head, some of which we confirm with classical FISH methods. Expression patterns of 100 genes in the brain and 50 genes in the body can be visualized interactively using TissUUmaps (*Solorzano et al., 2020*) at https://spatialfly.aertslab.org/.

## Results

### Creation of a comprehensive spatial dataset of the fly body and its head

In analogy to the FCA strategy (*Li et al., 2022*), we intended to map the spatial expression of a selected gene panel across the entire adult fly body and, in a separate set of experiments, across the adult head. Based on expression data from the FCA (*Li et al., 2022*; *Pech et al., 2024*) we selected 50 genes for the fly body samples, and 100 genes for the head samples given the large heterogeneity of neurons. These genes were carefully chosen to label the most important known cell types of the adult fly, and to include some unknown ones, suggested from the atlas data. They cover a wide range of expression levels, from high to very low expressed genes (*Supplementary file 1*, *Supplementary file 2*, *Figure 1—figure supplement 1*). Adult fly samples were frozen, sectioned, attached to slides, and fixed (see Methods), after which they were profiled using MC (*Figure 1a*). Using this workflow, we detected on average 190,622 mRNA molecules for the head samples (min = 101,548, max = 260,989), ranging from 56 (*Poxn*) to 23,439 (*trio*) mRNA molecules per gene (*Figure 1—figure supplement 2A, B*) and an average of 1.5 million mRNA molecules for the body samples (min = 1,448,593, max

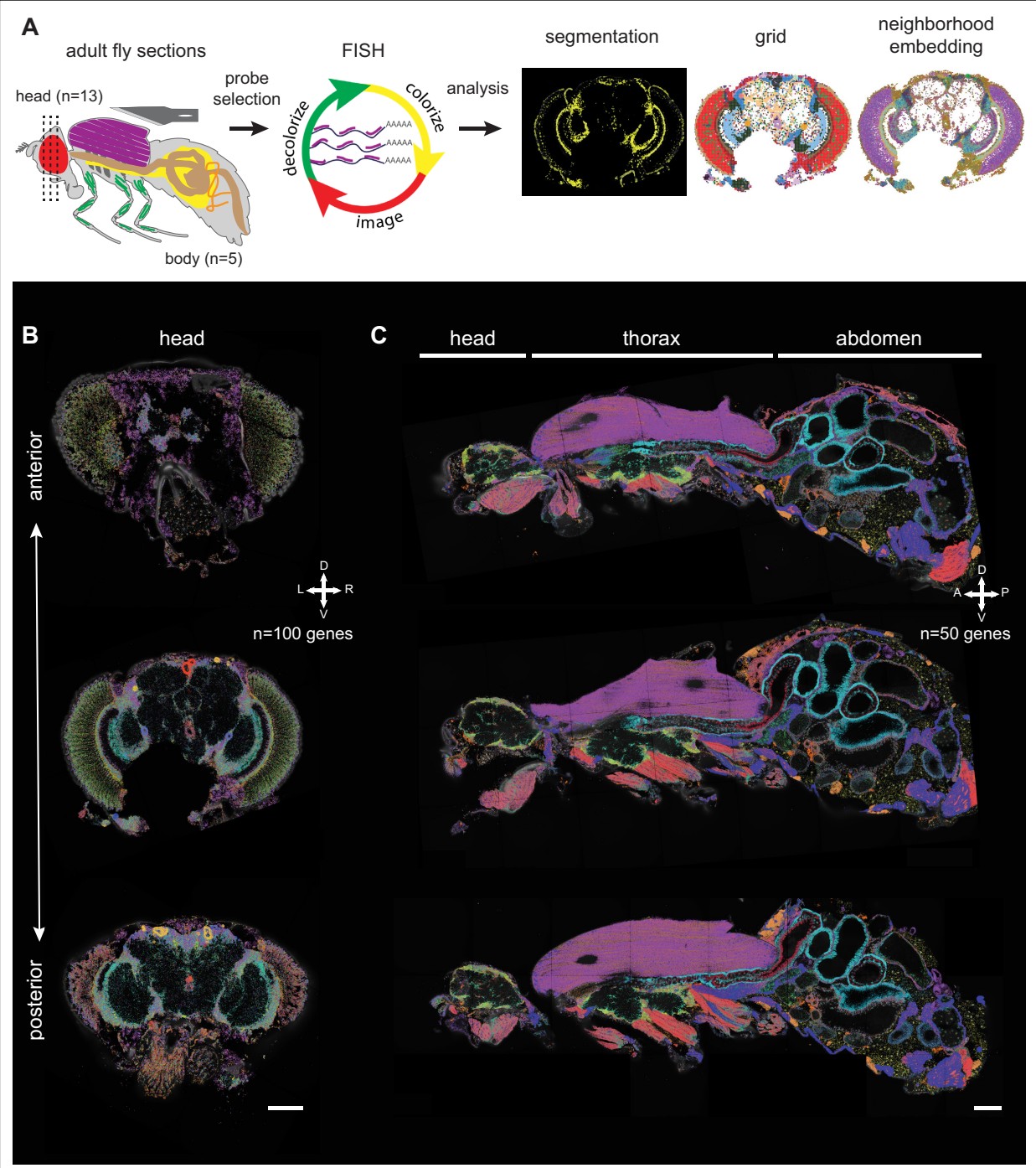

**Figure 1.** Principal workflow of spatial fly transcriptomics. (**A**) Overview of the spatial transcriptomics workflow: adult flies were sectioned, sections were analyzed with Molecular Cartography and data were annotated using cell segmentation, rasterization (i.e. grid), and neighborhood embedding (see Methods). (**B, C**) Three examples of adult head sections showing various positions in the brain along the anterior-posterior axis (**B**). Three examples of male whole-body sections were taken from the same male (**C**). mRNAs from each gene are represented in a different color. The combination of colors reveals the different cell types. Scale bars represent 100 μm. Background stain labels DAPI.

The online version of this article includes the following figure supplement(s) for figure 1:

**Figure supplement 1.** Expression of the here selected marker genes in single-cell datasets.

**Figure supplement 2.** Head samples.

**Figure supplement 3.** Body samples.

= 1,727,479), ranging from 112 (*MsR1*) to 510,619 (*Act88F*) mRNA molecules per gene (*Figure 1—figure supplement 3A, B*).

For the head samples, we aimed to image as many different neuronal populations as possible. Thus, we performed coronal sections at different depths in the head (13 sections, 12 animals, mixed sex, *Supplementary file 3*), along the anterior-posterior axis (*Figure 1B*, *Figure 1—figure supplement 2A, C*). For the body samples, sagittal sections were performed through the middle of the animal (5 sections from one male) (*Figure 1C*, *Figure 1—figure supplement 3A*). These sagittal sections show only small structural differences in the abdomen; overall they are very similar to each other, showing the reproducibility of our data (*Figure 1—figure supplement 3C*, see Methods).

## Spatial transcriptomics allows the identification of cell types in the body

The genes for the body datasets were selected to cover a wide range of different cell types. Neurons in the central and peripheral nervous system were identified using expression of *elav* and *Syt1* (*Figure 2A*). The glial marker *repo* shows the location of glial cells across the body (*Figure 2A*). The expression of different trypsin isoforms is unique to the digestive system. Interestingly, *α*- and *β-Trypsin* show distinct patterns, with *β-Trypsin* localized to the inner side of the gut, while *α-Trypsin* is more distal, suggesting a subcellular localization of these trypsin isoforms coding mRNAs to apical or basal regions of the gut enterocytes (*Figure 2A*). Using the fat body marker *AkhR* and the oenocyte marker *FASN2*, different populations of fat tissue and oenocytes were identified in the abdomen of the fly at the expected locations. The hemocyte marker *Hml* shows distinct local enrichments in the head, thorax, and abdomen (*Figure 2A*). In addition, *LanB1*, which codes for LamininB1, an important component of the extracellular matrix present around many tissues (*Yarnitzky and Volk, 1995*), is widely produced in different cell types and not only in hemocytes. While co-expression of *LanB1* with *Hml* in hemocytes is detected as reported (*Matsubayashi et al., 2017*) (15.5%) (*Figure 2A*), surprisingly most of *LanB1* overlaps with epithelial cells (*grh* (13.4%) and *pain* (28.0%)) or muscle cells (*Mlp84B* (40.9%) and *Mp20* (25.4%) (see Methods)). Next, we used *esg* to mark adult stem cell populations. The expression of *esg* in our sections is mostly limited to the gut, matching its reported expression in the intestinal stem cells (*Jiang and Edgar, 2011*) and the somatic cyst stem cells (*Sênos Demarco et al., 2022*; *Figure 2A*). Furthermore, *Mhc* and *sls* were used to label all muscle cells, while *TpnC4* and *Act88F* specifically label the IFMs (*Figure 2A*). Interestingly, in the IFMs we also detect expression of VGlut and other neuronal markers, consistent with glutamatergic neuromuscular junctions (*Schuster, 2006*; *Figure 2—figure supplement 1*). Finally, we used *grh, hth, svp,* and *pain* to identify epithelial cells and their subtypes (*Figure 2A*). Together, these data show that our spatial transcriptomics data can identify the spatial location of most known large classes of cell types of the adult fly, while also detecting unexpected subcellular mRNA localizations in some cell types.

## Molecular Cartography is highly specific

Most marker genes used in this study were selected based on FCA data to be highly specific for one particular cell type or present at one body location. Hence, their expression can be used to estimate the specificity of the mRNA detection method and determine false-positive rates of mRNA spots. For this specificity analysis, we chose to work with marker genes linked to muscle cells. We used the expression of *Act88F* and *TpnC4*, both of which are almost exclusively expressed in the IFMs (and possibly in muscles around the ejaculatory bulb) (*Fyrberg et al., 1983*; *Agianian et al., 2004*; *Sarov et al., 2016*). We segmented the flight muscles (*Figure 2A*) and calculated the percentage of marker genes detected in the segmented area compared to the entire imaged area. We find that 99.1% of *TpnC4* and 99.7% of *Act88F* mRNA spots are detected in the expected regions of all body sections (*Figure 2A*), with negligible off-target signal. This highlights the specificity of the transcriptomics method and the reliability of detected mRNA signals with MC.

## Building a spatial tissue atlas of the fly body

Cell types can often only be reliably identified by a combination of several marker genes. Therefore, we investigated co-expression signatures of known cell type marker genes in the spatial transcriptomics data. This may also help identify more rare cell types or specific cell states. To implement this idea on our spatial data, we computationally assigned every mRNA molecule to a 5 µm by 5 µm square

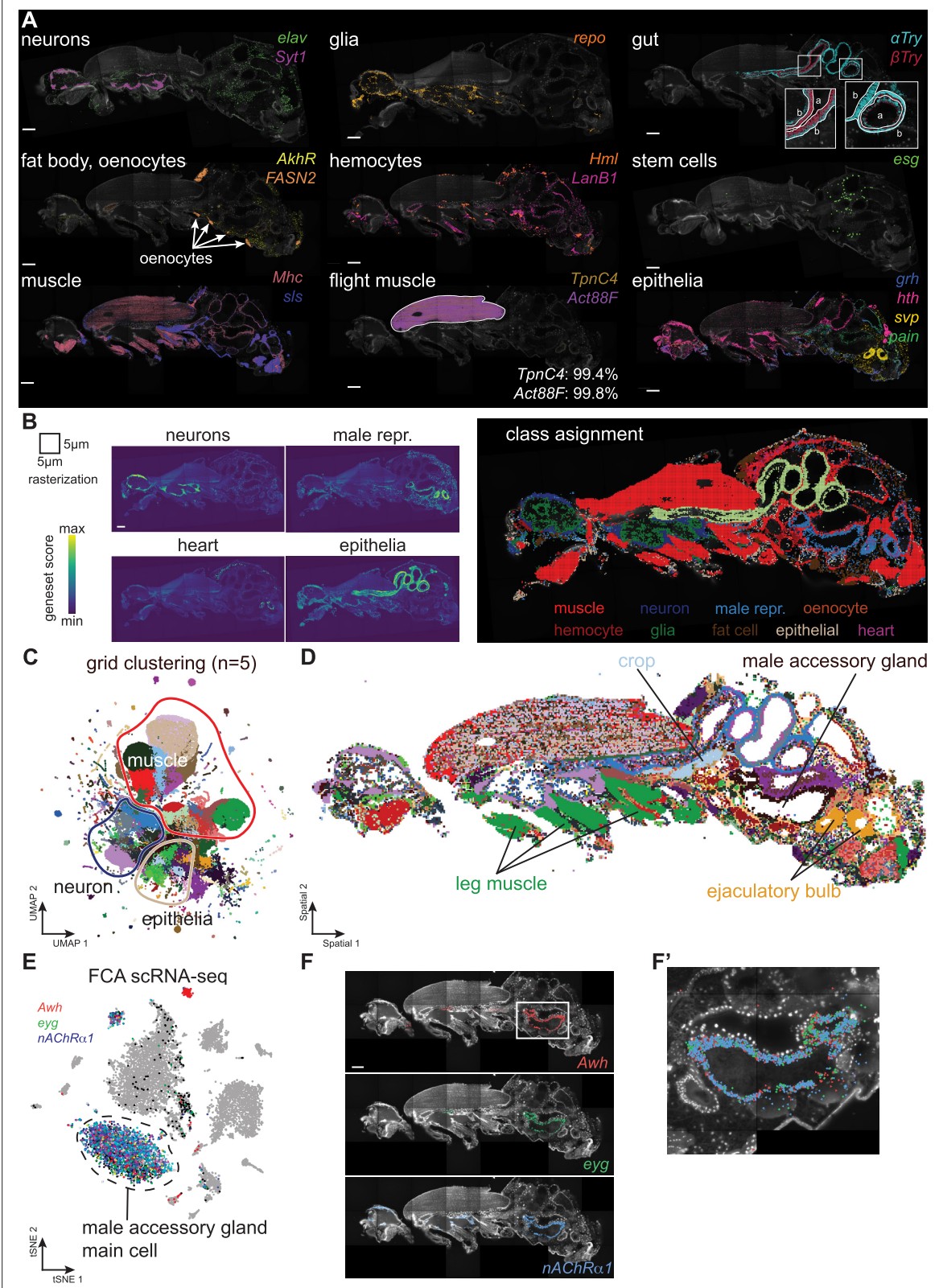

**Figure 2.** Adult body cell types. (**A**) Major cell types of adult males identified by marker genes. Scale bars represent 100 μm. Inset for gut shows zoom-ins of different regions (a: apical, b: basal). Inset for flight muscle, shows the percentage of marker gene molecules detected within the outlined area in the section shown. (**B**) Gene set scores for the main classes of cell types, quantified using 5 μm × 5 μm grid. Scale bar represents 100 μm. The class assignment shown on the right is based on the maximum score across classes. Genes used: neurons (*elav, Syt1, Sh, acj6, ey, VAChT, Gad1, VGlut,*

*Figure 2 continued on next page*

*Figure 2 continued*

nAChRα7); male reproductive system (*Awh, eyg, svp*); epithelia (*grh, αTry, βTry, hth*); heart (*tin, Hand*); muscle (*Mhc*); hemocyte (*Hml*); glia (*repo*); fat cells (*AkhR*); oenocytes (*FASN2*). (**C**) UMAP showing clustering of 5 µm × 5 µm grid spots. (**D**) Spatial location of grid clusters. (**E**) tSNE from male accessory glands from fly cell atlas (FCA) showing expression of marker genes for main gland cells. (**F, F'**) Molecular Cartography (MC) of the main gland cells marker genes highlights several defined populations of cells. Scale bar represents 100 µm. Detailed view shown in F'. Background stain labels DAPI.

The online version of this article includes the following figure supplement(s) for figure 2:

**Figure supplement 1.** Motor neuron markers in flight muscles.

**Figure supplement 2.** Comparison of body spatial datasets with body single-cell datasets.

**Figure supplement 3.** Colocalization of gene expression in the body datasets.

in a square-grid fashion and summed the signal, creating pseudo-bulk samples. All squares in this grid were then scored for mRNA signatures of different cell classes and assigned to the highest-scoring cell class (*Figure 2B*, see Methods). This signature-based method annotated all major cell types or tissues of the adult reliably, including muscles, neurons, glia, fat and epithelial cells, oenocytes, and the male reproductive organ (*Figure 2B*). In addition, cell populations with more sparse gene expression like the heart (combination of *Hand* and *tin*) or the male reproductive tract (combination of *Awh, eyg,* and *svp*) were newly identified. In conclusion, the major cell types of the adult fly could be localized using combined sets of marker genes identified by their spatial proximity.

As a next step, we aimed to assign cell-type locations without prior knowledge of their markers. To do so, we used all 132,642 different squares from each of the 5 body samples and performed clustering based on the counts of the 50 genes analyzed (see Methods). This led to the identification of 142 clusters that we visualized as a UMAP plot, separating muscle from neuronal and epithelial cells (*Figure 2C*). The spatial visualization of this unsupervised analysis of cell types confirmed our previous supervised annotation of cell types based on gene signatures (*Figure 2D*). In addition, it showed an unexpected presence of multiple clusters in the large flight muscle cells, suggesting subcellular mRNA patterns.

Finally, we looked at the expression of genes in one particular organ to identify its different cell types. In the FCA, we had previously identified *Awh, eyg,* and *nAChRα1* to label the male accessory gland main cells (*Figure 2E*). Our SRT data confirmed this co-expression, showing the labeling of the main gland cells in the abdomen (*Figure 2F and F'*).

Furthermore, we inspected whether co-expression of genes is maintained between SRT data and FCA snRNA-seq data, by calculating the correlation between genes in both datasets (see Methods). In general, gene-gene correlations between SRT and snRNA-seq match significantly ($r$=0.69, p<1e-100), although there are some small biases between the techniques (*Figure 2—figure supplement 2A, B*). For example, *Act88F* is detected at higher numbers in SRT, compared to snRNA-seq, which may be explained with differential mRNA stability in the cytoplasm, since snRNA-seq is largely detecting nuclear-located mRNAs and thus rather monitors the transcriptional activity of the cell. This indicates that in general SRT data corroborate results obtained from snRNA-seq. To study gene-gene relations in more detail, we devised an algorithm that calculates the colocalization of each gene with all other genes (*Figure 2—figure supplement 3A, B*, see Methods). This colocalization matrix was then clustered to find groups of co-expressed genes. These co-expression signatures match with markers of the major cell types (*Figure 2—figure supplement 3C*).

In conclusion, we present the first SRT dataset of the adult fly body, with high specificity and high reproducibility across body sections. We show how the SRT data can unambiguously identify the major cell types in both supervised and unsupervised techniques. Finally, SRT data can be used to describe gene-gene interactions.

## Subcellular localization of mRNA in muscle cells

Unsupervised clustering revealed the presence of multiple spatial niches in the muscles, especially in the IFMs (*Figure 2C and D*). Muscle cells form polynucleated syncytia, with more than 500 nuclei in one IFM cell (*Chaturvedi et al., 2017*; *Kaya-Çopur et al., 2021*). Thus, spatial niches could represent heterogenous nuclei or subcellular mRNA localization patterns, resulting from directed mRNA transport or anchoring (*Das et al., 2021*). To investigate this in more detail, we segmented the three

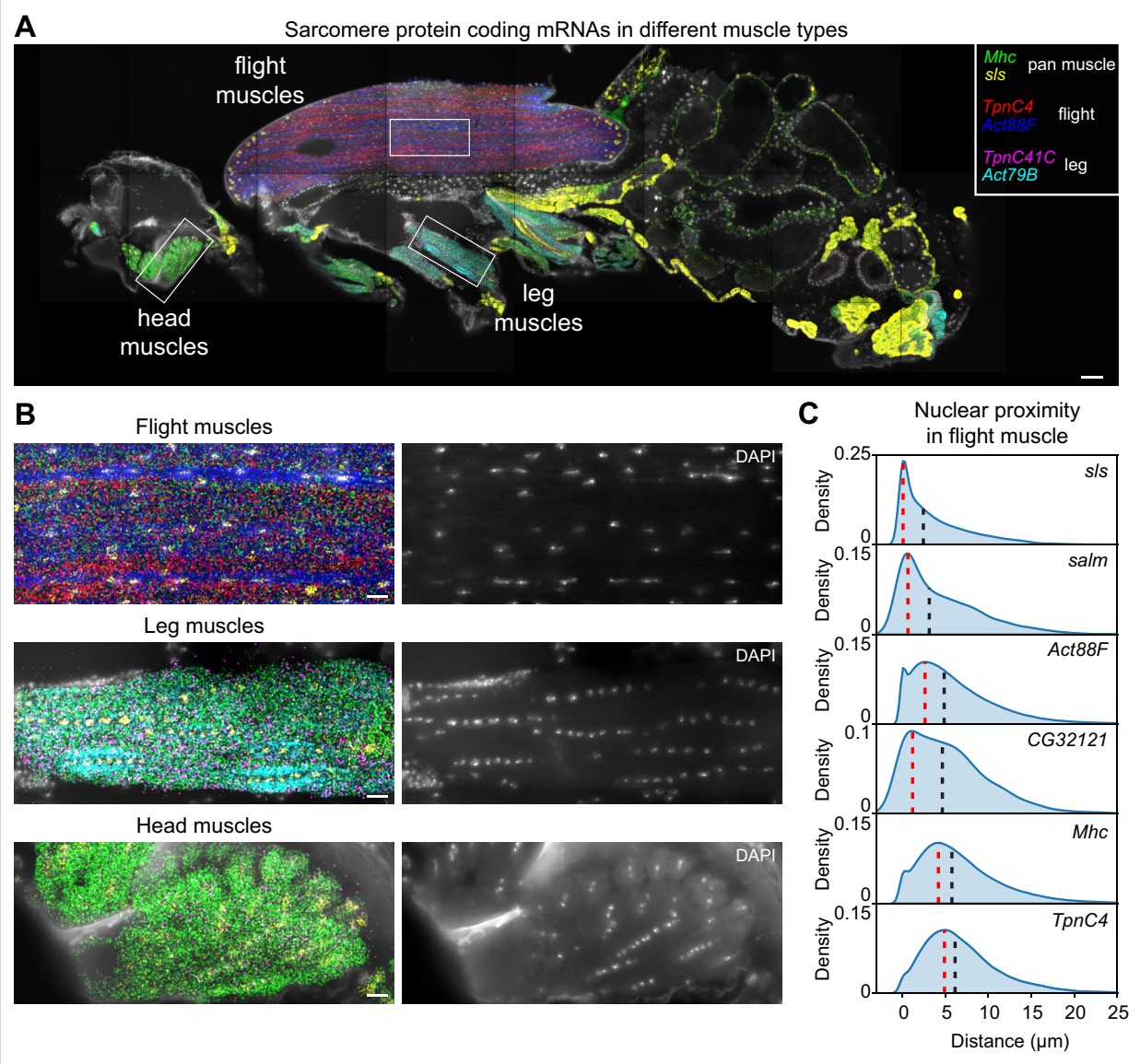

**Figure 3.** Molecular Cartography shows mRNA-specific nuclear enrichment. (**A**) Molecular Cartography (MC) visualization of marker genes of muscle subtypes. White boxes mark zoom-in regions shown in (**B**) and *Figure 3—figure supplement 1*. (**B**) Zoom-in on flight, leg, and head muscles. Left, labels are the same as in (**A**); right, DAPI-labeled nuclei. (**C**). Density plots show the distance of each mRNA molecule of the indicated genes to its nearest nucleus. Red dotted lines mark the peak density, and black dotted lines the median distance. Scale bars represent 50 μm in (**A**) and 10 μm in (**B**). Background stain labels DAPI.

The online version of this article includes the following figure supplement(s) for figure 3:

**Figure supplement 1.** Subcellular mRNA localization in leg and head muscles.

main types of adult muscles (head, leg, and indirect flight muscles) to look at spatial gene expression in them.

Amongst the 50 genes we had selected were six genes that code for sarcomere protein components, four of which are specifically expressed in different muscle groups: *Act88F* and *TpnC4* in IFMs, *Act79B*, and *TpnC41C* in leg muscles (*Figure 3A*). First, we investigated if mRNA species have different nuclear proximities by quantifying the distance of every mRNA molecule to its closest nucleus (see Methods, the localization precision of mRNAs is about 0.14 μm). This identified genes whose mRNAs are nucleus-enriched (*sls*) or nucleus-depleted (*TpnC4*) (*Figure 3B and C*, see also *Figure 4—figure supplement 1A*). We found a similar difference in leg and head muscles with *sls* staying in proximity or within the nuclei, in contrast to the nuclei-depleted *TpnC41C* (*Figure 3—figure supplement 1A*,

*B*). Previously, mRNA distributions in cultured mammalian skeletal muscles were linked to mRNA sizes, with large mRNAs spreading further from the nuclei than small mRNAs (*Pinheiro et al., 2021*). This does not seem to apply to our selected genes, as *sls* mRNAs are significantly longer (isoform lengths: 15,263–56,489 nt) compared to *TpnC4* (longest isoform: 859 nt) or *TpnC41C* mRNAs (longest isoform: 1,880 nt).

Furthermore, the nuclear enrichment of *sls* mRNA is not homogeneous across all nuclei, particularly in the IFMs, where *sls* mRNA forms large assemblies around nuclei close to the muscle-tendon junctions (*Figure 4A–C*). We confirmed these results by localizing *sls* mRNA using hybridization chain reaction technology (HCR-FISH) (*Choi et al., 2018*) in the thoraces from three adult males (*Figure 4D–F*, see Methods for details). We observed some small differences between MC and HCR, with *sls* mRNA enriched in broader regions around the muscle nuclei close to the tendons when detected with MC. This may be due to differences in fixation timing (pre-sectioning fixation for HCR, post-sectioning fixation for MC) or in image generation (direct imaging of hybridization events in HCR vs detecting, localizing, and combining multiplexed hybridization events in MC). Similarly enriched localizations of mRNA in muscles were reported in mammals close to the muscle-tendon junctions and the neuro-muscular junctions (*Dix and Eisenberg, 1990*; *Sanes et al., 1991*). In mammals, this local enrichment correlates with an accumulation of nuclei (*Bruusgaard et al., 2003*), however, we do not detect a significant correlation between the *sls* enrichment at the IFM ends and the nuclear DAPI signal (*Figure 4—figure supplement 1*) and thus speculate that the terminal nuclei express higher levels of *sls* RNA than the central muscle nuclei.

In addition to nucleus-enriched or -depleted mRNA species, MC revealed mRNAs coding for several sarcomere proteins to be concentrated in longitudinal stripes in the muscles. Most prominently in the IFMs, *Act88F* mRNA is enriched in the inter-myofibril space, where most nuclei, endoplasmic reticulum, and Golgi are located, while *TpnC4* and *Mhc* mRNAs appear enriched in complementary stripes at the location of the myofibrils and mitochondria (*Figure 5A and B*). This suggests a spatial subdivision of the flight muscles that correlates with their intracellular architecture (*Avellaneda et al., 2021*; *Luis and Schnorrer, 2021*). A similar division was found in leg muscles, with central stripes of *Act79B* mRNA close to the nuclei while *Mhc* and *TpnC41C* mRNAs were enriched in the myofibril regions (*Figure 3—figure supplement 1B*). To validate these observations, we performed HCR-FISH. This partially confirmed the anticorrelated expression patterns between *Act88F* and *TpnC4* in IFMs: *Act88F* mRNA was found enriched in stripes along the rows of nuclei, consistent with the MC data. However, *TpnC4* mRNA was also found enriched in a few cases around nuclei (*Figure 5C, D*, *Figure 5—figure supplement 1*). Additionally, HCR-FISH detected that *Mhc* mRNAs are enriched in proximity to the nuclei and not in proximity to the myofibrils as suggested by MC (*Figure 5E and F*). Overall, these data suggest that the main observations obtained here with MC are reliable, however, a systematic validation of SRT techniques with classical FISH is recommended.

To conclude, we found three main types of mRNA localization in the IFMs: nuclei-enriched patterns, complementary striped bands, and concentrations at the terminal nuclei, close to the muscle attachment sites. To our knowledge, none of these patterns had been identified before.

## Spatial transcriptomics allows the localization of cell types in the head and brain

To investigate the localization of cell types in the adult *Drosophila* head, we investigated the 13 sections of different fly heads sectioned across the longitudinal direction, covering most regions of the fly head. The diversity between sections originating from different brain regions is represented by lower correlations between the samples (*Figure 1—figure supplement 2C*). Using the marker genes *para* (neurons), *alrm* (astrocytes), and *ninaC* (photoreceptors), we were able to annotate different classes of cells (*Figure 6A*). *ninaE*, a second photoreceptor marker was too highly expressed, leading to optical crowding, and was removed from the analysis.

First, we made use of our annotated scRNA-seq atlas of the fly brain (*Davie et al., 2018*; *Li et al., 2022*; *Pech et al., 2024*), using colocalization of marker genes to identify cell types with known anatomical locations. We used the expression of *C15*, *acj6*, *Oaz*, *caup*, and *unpg*, markers for olfactory projection neurons (OPNs). OPNs relay information from the antennal lobe to downstream processing centers, including the mushroom body and the lateral horn. The co-expression of OPN markers is only detected in the most anterior samples, as expected (*Figure 6B*). Next, we looked at *repo*, *moody*,

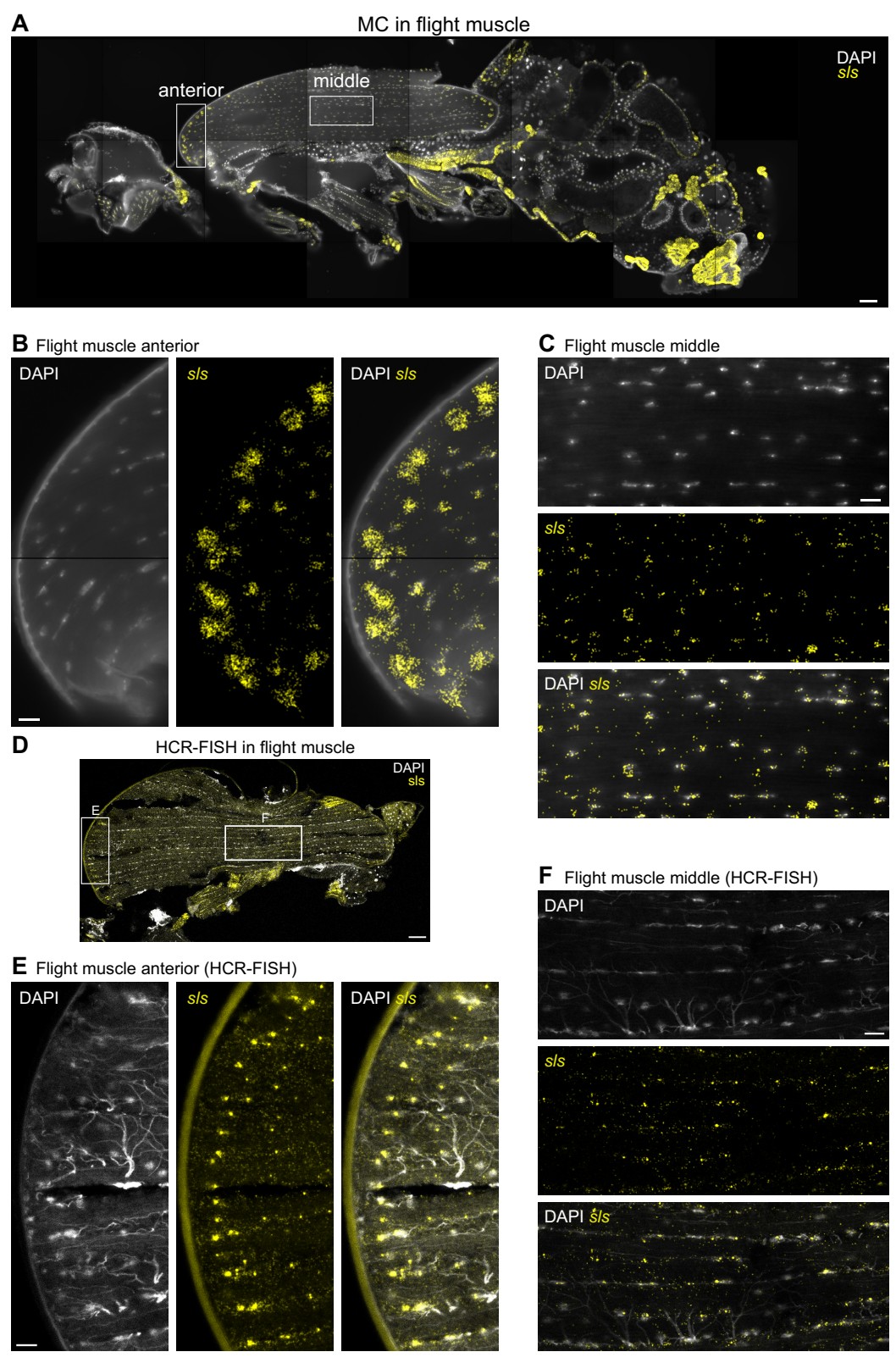

**Figure 4.** *sls* mRNA shows nuclear enrichment with increased concentration close to muscle-tendon junctions. (**A**) Molecular Cartography visualization of *sls* mRNA (yellow) as pan muscle maker (same section as in **Figure 3**). White boxes mark zoom-in regions shown in (**B**) and (**C**). (**B, C**) Zoom-in on indirect flight muscles showing colocalization of *sls* mRNAs and DAPI stained nuclei in anterior (**B**) or central (**C**) regions of the flight muscle. (**D**) Hybridization

*Figure 4 continued on next page*

*Figure 4 continued*

chain reaction-fluorescence in situ hybridization (HCR-FISH) imaging of *sls* mRNA in an adult thorax. White boxes mark zoom-in regions shown in (**E**) and (**F**). (**E, F**) Zoom-ins on flight muscle at anterior (**E**) or middle (**F**) regions. Note the sls mRNA and DAPI co-localization (tracheal cells show background stain in the DAPI channel). Scale bars represent 50 µm in (**A**) and (**D**), and 10 µm in (**B**), (**C**), (**E**), and (**F**). Background stain labels DAPI.

The online version of this article includes the following figure supplement(s) for figure 4:

**Figure supplement 1.** Nuclear and *sls* mRNA localization.

---

*Indy*, *Vmat*, all of which are markers for perineurial glia that form the blood-brain barrier (BBB), and found them expressed at the periphery of the brain, consistent with the known location of the BBB (*Figure 6C*).

We compared gene-gene relationships, by calculating gene-gene correlations in MC and the snRNA-seq brain data (*Figure 6—figure supplement 1A, B*). In general, we find a strong match ($r=0.68$), however, there are also several mismatches. While *pros-dati* co-expression is conserved between both modalities (*Figure 6—figure supplement 1C*), co-expression of several other gene pairs is only detected in one modality (*Figure 6—figure supplement 1D, E*). For example, we only found widespread expression of Vmat in the BBB (stronger near the optic lobe), but little or no expression in the central brain, where most dopaminergic neurons should be located (*Figure 6—figure supplement 1E*). Additionally, *Vmat* expression does neither overlap with DAPI-stained nuclei nor with *DAT* (a marker for dopaminergic neurons), suggesting an mRNA transport mechanism to locate selected mRNAs away from the cell body (*Figure 6—figure supplement 1E*). Alternatively, glial nuclei, which are much smaller than neuronal nuclei (*Mu et al., 2021*), might not all be detected with our methodology. Similarly, the neuropeptides *Ilp2* and *Pdf* show only a weak overlap with the DAPI staining, instead forming circular patterns (*Figure 6—figure supplement 1F*). Thus, MC can recapitulate the location of known cell types in the fly brain by using established marker genes and additionally identify expression at novel locations for some of these.

Compared with other organs, the cell type diversity of the fly brain is extremely complex. A key challenge in scRNA-seq is the annotation of clusters to cell types. In our scRNA-seq dataset, there are 188 distinct clusters, of which only 83 are annotated today. Recent efforts to map the fly brain through EM and connectomics studies have identified 8453 morphological cell types (*Dorkenwald et al., 2024*; *Schlegel et al., 2024*). SRT can provide a bridge between scRNA-seq studies and morphology-based studies. As such, it becomes possible to annotate unknown clusters based on the spatial localization of their marker genes. To pave the way toward such integration, we focused on unannotated clusters in our brain dataset. It is important to note that current neuronal nomenclature is based on neuropils (axons and dendrites) and not on the location of the neuronal nuclei. Therefore, we used the established neuropil nomenclature to describe the location of the detected nuclei, but this does not necessarily mean that the axons of these cells also project there.

To identify the location of uncharacterized clusters from the FCA in the brain, we started with Fer1 expression, as it is a strong marker for cluster 120 in our snRNA dataset (*Li et al., 2022*). Using our MC data, we found that these cells are located in the central brain, near the ventral gnathal ganglion and saddle, and the superior intermediate protocerebrum (*Figure 6D*). Furthermore, we show that *AstC* expression (cluster 122) is limited to the dorsal part of the central brain, consistent with its expression in DN1 and DN3 neurons (*Díaz et al., 2019*; *Figure 6D*). Next, we investigated the expression of *otp* (clusters 30 and 62),which marks nuclei close to the lateral horn and the superior neuropils (*Figure 6E*). Finally, we determined the expression of *vg* (cluster 20) to be limited to the boundary region between the optic lobe (OL) and the central brain (CB) (*Figure 6E*). These results confirm that combining snRNA-seq with SRT technologies can indeed identify defined locations of formerly uncharacterized cell clusters in the fly brain.

Next, we wanted to find genes enriched to specific spatial locations in our brain MC data. To do so, we assigned mRNA molecules to spots using a similar 5 µm resolution grid as done above for the body data. By performing unsupervised clustering using all mRNA localizations from all datasets together, we identified 23 clusters (*Figure 6F*). Different slices display a different cluster composition, consistent with our aim for sectioning along the brain's A-P axis (*Figure 6—figure supplement 1G*). All neuronal clusters were selected using *para* expression and annotated to either optic lobe (OL), central brain (CB), or retina (photoreceptor, PR) based on their location. By performing differential

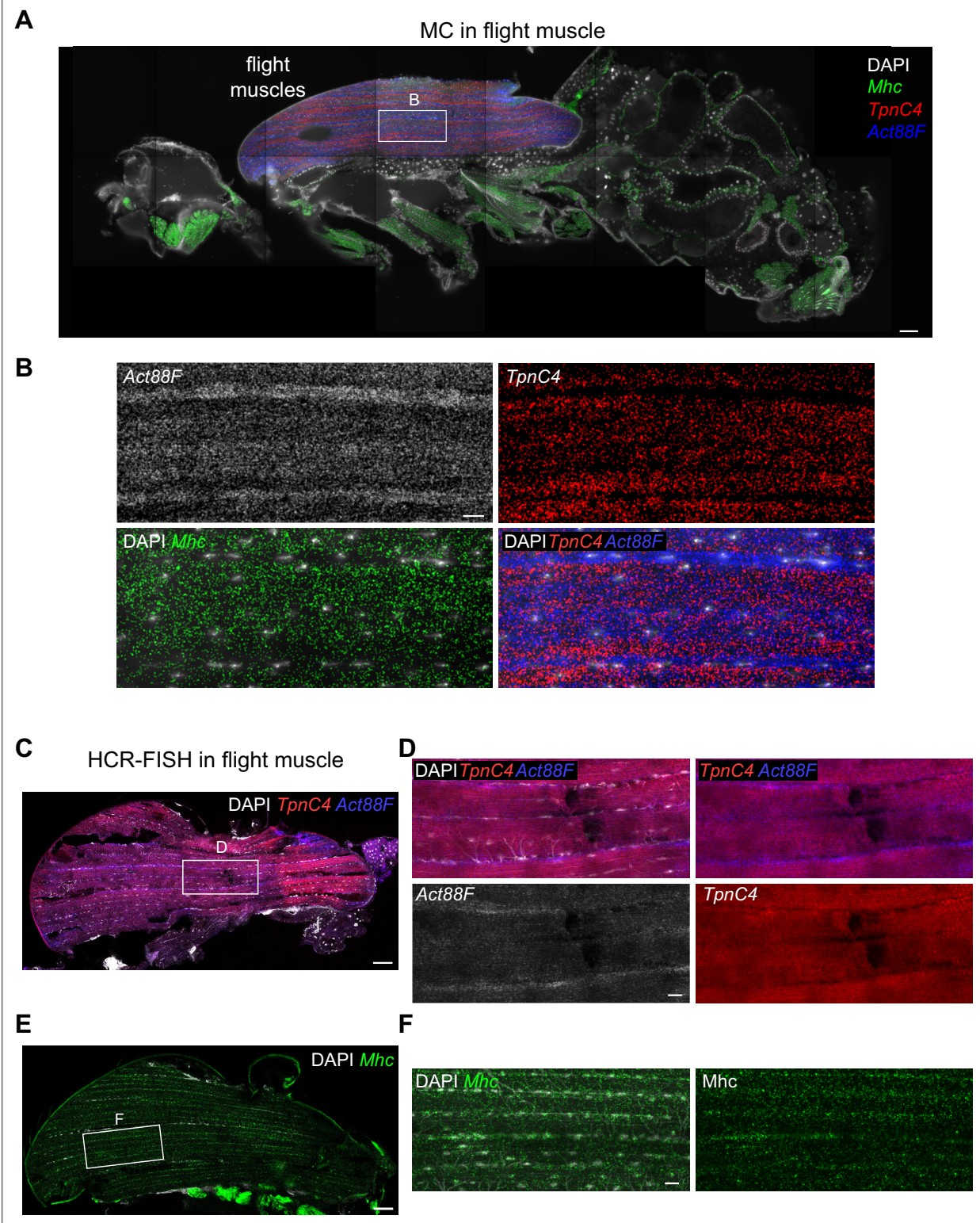

**Figure 5.** *TpnC4*, *Act88F*, and *Mhc* mRNAs flight muscle patterning. (**A**) Molecular Cartography visualization of *TpnC4*, *Act88F*, and *Mhc* mRNAs on an adult fly section (same section as in *Figure 3*). The white box marks the zoom-in region shown in (**B**). (**B**) Zoom-in on the flight muscle showing the striped patterns of *TpnC4*, *Act88F* (blue in overlay), and *Mhc* mRNAs in the indirect flight muscle. (**C**) Hybridization chain reaction-fluorescence in situ hybridization (HCR-FISH) of *TpnC4* and *Act88F* mRNAs in an adult thorax. The white box marks zoom-in the region shown in (**D**). (**D**) Zoom-in on the flight muscle of HCR-FISH labeling *TpnC4* and *Act88F* mRNAs (same section as in *Figure 4D*). (**E**) HCR-FISH of *Mhc* mRNAs in an adult thorax. The white

*Figure 5 continued on next page*

*Figure 5 continued*

box marks the zoom-in region shown in (**F**). (**F**) Zoom-in on the flight muscle of HCR-FISH labeling *Mhc* mRNAs. Scale bars represent 50 µm in (**A**), (**C**), (**E**). and 10 µm in (**B**), (**D**), (**F**). Background stain labels DAPI.

The online version of this article includes the following figure supplement(s) for figure 5:

**Figure supplement 1.** Variation in mRNA localization using Hybridization chain reaction-fluorescence in situ hybridization (HCR-FISH).

expression between these domains, we identified spatially relevant genes (*Figure 6G*). As expected, photoreceptor genes such as *ninaC* and *lz* were strongly enriched in the retina. While many neuronal genes show similar expression between the OL and CB, *pros* (CB) and *scro* (OL) expression creates a strong boundary clearly demarcating the two anatomical regions (*Figure 6G and H*), corroborating earlier findings (*Davie et al., 2018*; *Yoo et al., 2020*). This shows that SRT alone can be used to identify specific markers for the major brain cell types or regional domains, without any prior knowledge.

Previously, it was shown that some scRNA-seq clusters in the OL could be divided into ventral *Wnt4* and dorsal *Wnt10* positive subclusters (*Han et al., 2020*; *Özel et al., 2021*). Using *Wnt4* and *Wnt10* mRNA localizations we confirmed these findings and showed that, although expression is sparse, *Wnt4* and *Wnt10* mRNAs are strongly enriched in the ventral and dorsal regions of the brain, respectively (*Figure 6I*).

Altogether, our MC data of the fly brain allows the identification of clusters and cell types. We also showed how SRT can be used to identify and localize unknown clusters of scRNA data, serving as a potential link between morphology data and single-cell data. Finally, SRT can be used for the discovery and confirmation of regional marker genes by performing region-based differential expression.

## Automated cell type annotation of SRT using scRNA-seq data

One of the key prospects for SRT, is to be able to identify unlabeled clusters and infer their spatial location. This requires the integration of SRT data with scRNA-seq data. Multiple methods have been developed for this purpose so far, including Tangram (*Biancalani et al., 2021*) and SpaGE (*Abdelaal et al., 2020*).

Here, we used and compared three different approaches to represent SRT data: 5 µm-spaced grid rasterization, neighborhood embedding, and nuclei segmentation (*Figure 7A*). The grid and neighborhood embedding are both spatially unaware of the cell's location. In the grid method, automatic segmentation takes place over the entire sample by grouping mRNA molecules together every 5 µm, while neighborhood embedding is segmentation-free and models every mRNA molecule independently. This makes it possible to also visualize spatial patterns at subcellular resolution, but leads to a very large dataset and heavy computational load (*Partel and Wählby, 2021*). While the grid-based approach can be run in several hours, neighborhood embedding takes several days. Finally, we segmented the nuclei and assigned mRNA molecules to each nucleus. While this is the most intuitive method, several challenges occur in the fly head. To start, different densities of nuclei require imaging with different parameters to visualize both sparse and dense nuclei regions. In addition, the dense nuclei regions at the edge of the OL and CB are very difficult to segment with normal imaging techniques, with a strong overlap of cells and different cell types (*Figure 7B*).

To compare label transfer methods we applied lasso regression, Tangram, and SpaGE. We found that regression-based integration in the grid method and SpaGE in the neighborhood embedding method can match the blood-brain barrier and the chiasm glia in the optic lobe very well, together with astrocytes and ensheathing glia in the central brain (*Figure 7C*). In contrast, the nuclei segmentation method performs poorly for matching glial subtypes, since most of the glial nuclei are not detected, leading to the removal of most glial mRNAs (*Figure 7C*). Therefore, we also ran Tangram on the grid-quantification, leading to the detection of different glial types (*Figure 7—figure supplement 1A–D*). This shows a disadvantage of nuclei-aware methods compared to methods that take all mRNA spots into account.

In the OL, all methods can accurately identify neurons from the different layers. In the CB, both the grid-based and the segmentation methods are able to locate the peptidergic neurons (insulin-producing cells, Pdf-neurons) in the correct location, while neighborhood embedding fails to retrieve these (*Figure 7D and E*). However, Tangram also labels some non-peptidergic segmented nuclei as insulin-producing unless thresholds for mapping scores are manually adjusted to reflect the different

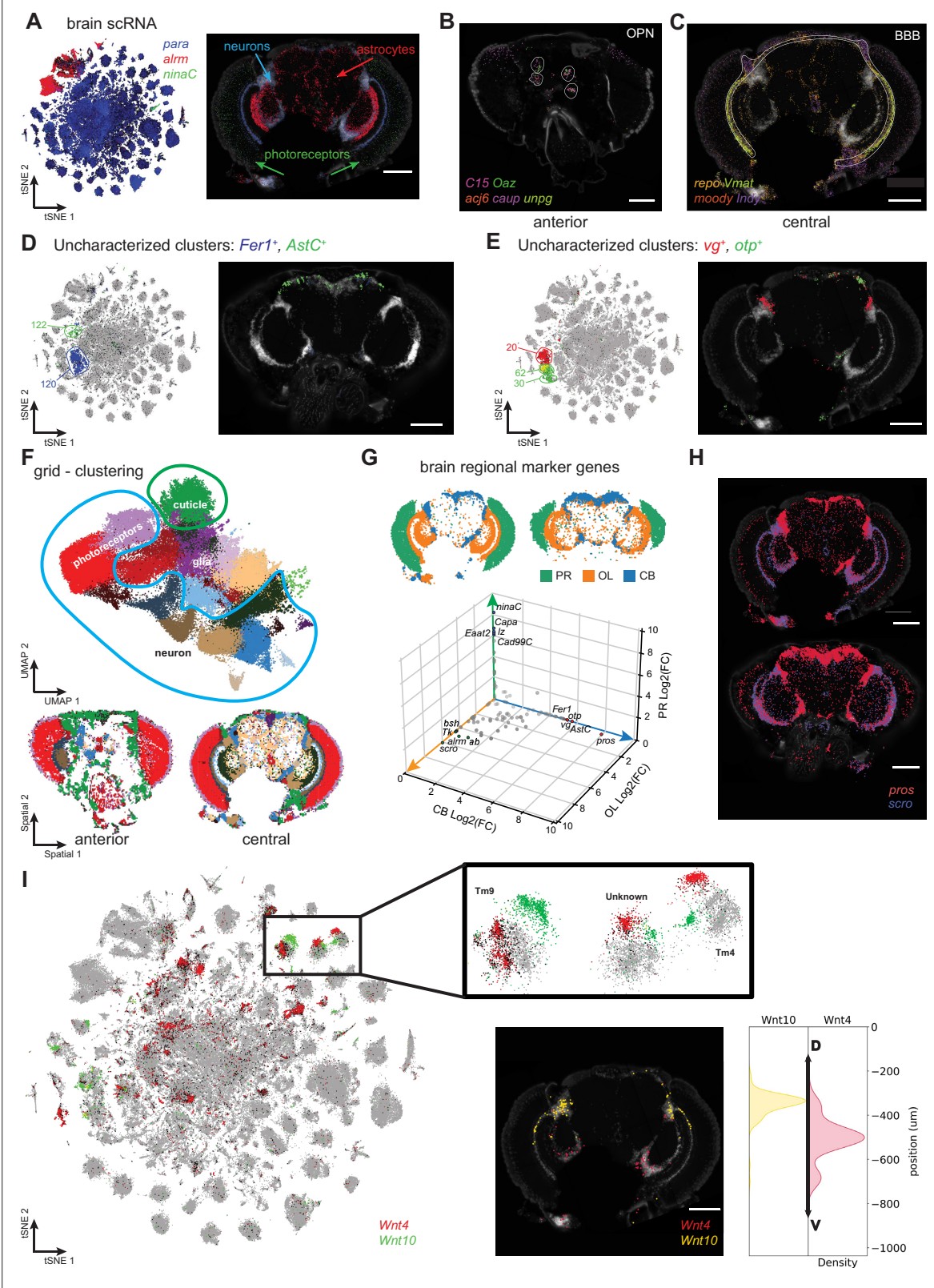

**Figure 6.** Adult head cell types. (**A**) tSNE showing expression of photoreceptor (*ninaC*), neuronal (*para*) and glial (*repo*) markers (left). Molecular Cartography of the same marker genes (right). (**B, C**) Molecular Cartography of marker genes for olfactory projection neurons (OPN), in an anterior head slice (**B**) and of perineurial glia of the blood-brain barrier (BBB) in a more central brain slice (**C**). (**D, E**) Using Molecular Cartography to localize uncharacterized clusters found in scRNA-seq data. (**F**) UMAP showing clustering of 5 µm × 5 µm grid spots (top). Spatial location of grid clusters in the

*Figure 6 continued on next page*

*Figure 6 continued*

brain (bottom). (**G**) Differential expression of central brain (CB), optic lobe (OL) and photoreceptor regions (PR). (**H**) Molecular Cartography of *pros* and *scro* in the brain. (**I**) tSNE showing split in optic lobe clusters by expression of *Wnt4* and *Wnt10*. Insert shows Molecular Cartography of *Wnt4* and *Wnt10*, spatially localized in ventral and dorsal brain regions, respectively. Scale bars represent 100 µm. Background stain labels DAPI.

The online version of this article includes the following figure supplement(s) for figure 6:

**Figure supplement 1.** Comparison of head spatial datasets with brain single-cell datasets.

proportions of cell types that are present in the MC data (*Figure 7—figure supplement 1E*). In a more posterior CB sample, nuclei segmentation and grid methods are able to identify the Kenyon cells of the mushroom body. These are also found in the neighborhood embedding, but the signal is noisier (*Figure 7E*). Neighborhood embedding likely performs poorly on cell types that are determined by high expression of only 1 gene (e.g. neuropeptides), since one gene only has a marginal effect on the neighborhood.

Finally, we investigated how the three approaches perform in linking uncharacterized scRNA-seq clusters to spatial locations. The grid and neighborhood embedding methods spatially localized two uncharacterized clusters, while the nuclei segmentation method spatially localized four clusters (*Figure 7F*). Cluster 20, which was marked by *vg* expression, is found by each technique (*Figures 6E and 7F*), same as cluster 79. Cluster 79 is located to the outside of the retina, fitting with high expression of *Cpr72Ec*, a structural protein for the eye lens, which is expressed by interommatidial pigment cells (*Stahl et al., 2017*). In contrast to the other methods, Tangram detected a ventral-dorsal division in the retina, with cluster 15 matching to the dorsal part. This mapping is driven by the expression of *mirr*, a gene known to be expressed in the dorsal half of the eye (*McNeill et al., 1997*; *Figure 7—figure supplement 1F*). Tangram also maps cluster 69, mostly by expression of *caup*. Like its Iroquouis family member *mirr*, *caup* is also detected in the dorsal half of the eye, however, it shows additional expression in various locations in the central brain (*Figure 7—figure supplement 1G*). Cluster 69 represents multiple subclusters in scRNA-seq data, and similarly, it maps to multiple locations in the MC data. A larger gene panel will help to dissect and locate the subclusters in more detail in the future.

To conclude, we found that using a 5 µm by 5 µm grid leads to similar clustering results as performing a computationally heavy neighborhood embedding analysis. However, the neighborhood embedding does increase the resolution to single mRNA spots, leading to a better visualization and a higher spatial accuracy. The nuclei segmentation method is limited in the fly brain by the inability of the DAPI signal to resolve overlapping individual nuclei in very densely packed brain regions. While Tangram was the best method to identify unknown clusters, it also led to false-positive labeling of multiple cell types, leading to the necessity of accurate thresholding. We find that using a simple regression model works as well or even better as designated methods when marker genes are used, while running much faster. The designated methods might show improved performance when using larger gene panels. Therefore, we suggest relying on a consensus of different methods to optimally integrate scRNA-seq data with SRT.

## Discussion

Here, we present the first SRT dataset of the adult fly brain and body using a highly multiplexed single molecular in situ hybridization technique Molecular Cartography (MC). Most of the published classical mRNA in situ hybridizations (*Tomancak et al., 2002*; *Lécuyer et al., 2007*; *Jambor et al., 2015*), as well as a recent high throughput FISH study used the more easily accessible fly embryos (*Wang et al., 2022*), We expanded SRT here to the more complex anatomy of the adult fly, which requires sectioning and large data sampling, both at the microscope and at the computer. Using a 50-genes panel in the body, all major cell types in the adult male fly were identified. Furthermore, the expression of key markers was sufficient to locate rare cell populations in the adult body. For example, expression of *Hml* showed the location of resident hemocytes, while *esg* was used to mark stem cells in the reproductive and intestinal tracts. We further showed a high concordance with published snRNA-seq data using several measures of co-expression (*Li et al., 2022*).

While SRT based on smFISH in mice (*Yao et al., 2023*) and humans (*Fang et al., 2022*) is currently limited to a single tissue, in flies both the brain and the whole-body can be sliced and imaged, allowing

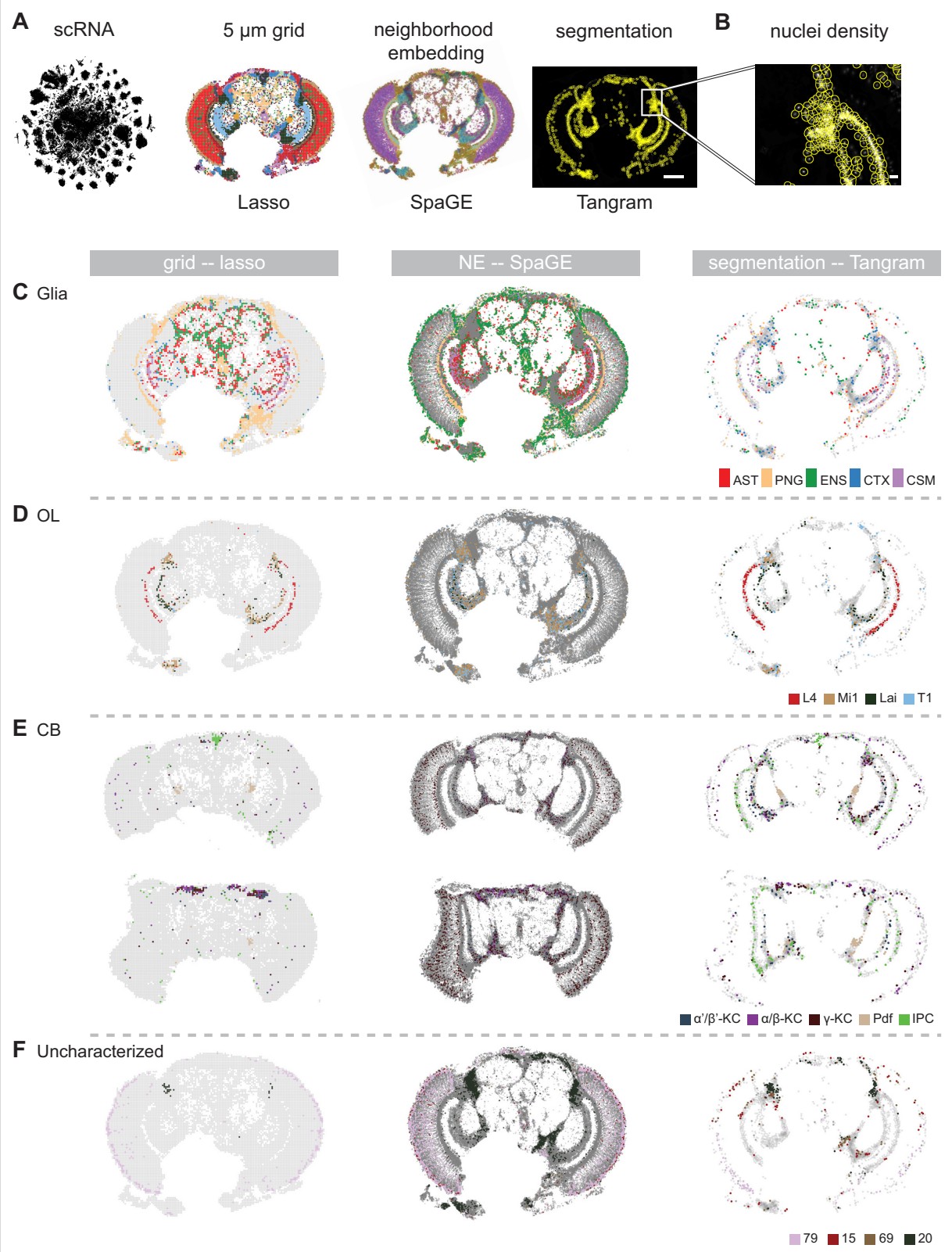

**Figure 7.** Comparison of different techniques for annotating the adult head samples. (**A**) Overview of different spatial analysis methods which were used to annotate Molecular Cartography with labels from single-cell RNA-seq: grid-based, neighborhood embedding and nuclei segmentation. Scale bar represents 100 μm. (**B**) Zoom-in on a high-density region with corresponding segmented nuclei. Scale bar represents 10 μm. (**C-F**) Comparison of annotation of spatial data with single-cell RNA-seq for three different quantification methods. Grid-based squares/neighborhoods/nuclei are colored

*Figure 7 continued on next page*

*Figure 7 continued*

based on matching single-cell clusters for (**C**) glia, (**D**) optic lobe, (**E**) central brain, and (**F**) four uncharacterized clusters. In (**E**) two brain slices are shown at different depths: central (top) and posterior (bottom). NE: neighborhood embedding.

The online version of this article includes the following figure supplement(s) for figure 7:

**Figure supplement 1.** Tangram on grid-based quantification.

an unprecedented whole body view. This enabled us to investigate the specificity of marker genes in the context of all tissues. We found that many markers are indeed highly specific to certain cell types, as we quantified for *Act88F* and *TpnC4* for the flight muscles (>99% specificity), which is consistent with previous benchmarks of MC in human and mouse cells (*Groiss et al., 2021*). Still, some marker genes showed additional unexpected expression (e.g. expression of *tin* and *TpnC4* in the ejaculatory bulb muscles), which was not detected in snRNA-seq. Additionally, we observed widespread expression of *LanB1* in various cell types, beyond just hemocytes. As such, SRT can provide a more sensitive readout, including the spatial localization, and may inspire which other tissues might be interesting to investigate in the future when studying a particular gene function.

In the head, we used a 100-genes panel and showed that this selection is sufficient to identify the major classes of cell types in the brain, including some very specific neuronal cell types like peptidergic neurons, Kenyon cells, and different OL neurons. Furthermore, by using informative markers enriched in previously uncharacterized clusters from single-cell RNA-seq data, we were able to pinpoint the nuclei in the fly brain that express these markers. This allowed us to map a series of previously uncharacterized cell clusters to specific brain regions. The next challenge lies in connecting these nuclei to the neuronal connectome using connectomics studies (*Schlegel et al., 2024*), in order to unambiguously identify the individual cell types. Indeed, in *Drosophila*, neuronal nomenclature and cell typing focus on projections, instead of nuclei localization. We believe that future large-scale SRT data can serve as a bridge between different modalities, similar to the Rosetta stone, to link scRNA-seq atlases with spatial information, including connectomics in the brain.

Performing SRT on the adult head led to two additional challenges. The first issue is technical, since the small area of the head made it difficult to achieve homogeneous adhesion to the slides, leading to significant deformation of the samples. Second, the small and dense *Drosophila* brain nuclei are hard to segment (average size 5 cubic micrometers *Mu et al., 2021*). A possible solution would be to use SRT-compatible expansion microscopy (*Fan et al., 2023*; *Pownall et al., 2023*; *Steib et al., 2022*). Alternatively, imaging techniques that allow super-resolution like DNA-PAINT, could help to achieve higher resolution (sub 5 nm resolution) (*Agasti et al., 2017*; *Schueder et al., 2023*). Third, apart from their small size, most *Drosophila* neuronal nuclei are found in dense regions, with significant overlap in 2D images. Thus, 3D imaging could be used to improve nuclei segmentation (*Pachitariu and Stringer, 2022*), but still leaves open the problem of overlapping neuronal projections.

Therefore, we explored several computational techniques to integrate SRT with single-cell RNA-seq, including grid-rasterization, nuclei-segmentation and neighborhood embedding. Given the dense packing of the small nuclei in the fly brain, and the likely presence of mRNA transport away from the nucleus of origin, nuclei-segmentation leads to a loss of information compared to the other nuclei-unaware techniques. We further found that the mapping of some clusters is dominated by the presence of strong marker genes. Therefore, we predict strong improvements when using larger gene panels. Currently, sequencing-based methods such as Stereo-seq hold an advantage in number of genes that can be profiled (*Wang et al., 2022*), but smFISH-based approaches are scaling rapidly and will soon allow to routinely image 1000s of genes at higher resolutions (*Fang et al., 2022*).

The most exciting advantage of SRT compared to single-cell techniques applied to cells or nuclei in suspension, is the possibility to study mRNA localization within the tissue-context. It was previously reported in the embryo that large numbers of mRNAs are non-homogeneously distributed and thus likely transported (*Jambor et al., 2015*; *Lécuyer et al., 2007*; *Tomancak et al., 2002*). In our SRT data, we found localized mRNAs in at least three different cell types. First, we found that different trypsin isoforms located in opposite apical-basal locations in enterocytes of the adult gut. Such apical-basal mRNA transport processes are well described in the embryonic epidermis (*Bullock et al., 2006*), however, to our knowledge not yet known in the adult fly gut epidermis, where secretion of the enzymes at the correct side of the cell should be critical.

Second, in the brain, we identified localized patterns of mRNAs coding for the neuropeptides Pdf and Ilp2. These mRNAs are likely transported into axons or dendrites, away from the cell body, a process often critical for brain development and function (*Holt and Bullock, 2009*). As both genes are important regulators of adult physiology, the identified localization patterns may inspire future studies.

Third, in muscle cells, SRT revealed different patterns of mRNA distributions within muscle fibers that suggest both mRNA transport and local synthesis in a subset of nuclei. Most prominently, *sls* mRNA remains close to the nuclei, whereas some muscle regions display alternating striped patterns of *Act88F* and *TpnC4* mRNAs that recapitulate the functional specializations within the muscle fibers, with myofibrils, mitochondria, and ER being present at defined locations (*Luis and Schnorrer, 2021*). It has been proposed that ribosomes are concentrated at particular locations in mammalian muscles to build translational hotspots (*Denes et al., 2021*; *Lewis et al., 2018*; *Rudolph et al., 2019*). Whether such concentrations exist in fly muscles is not known, and it will require future investigations to identify the localization mechanisms and the functional significance of localized mRNAs concentrations. Finally, we propose the existence of heterogenous nuclei in the IFMs, with terminal nuclei facing the muscle-tendon attachment sites showing an enrichment of *sls* mRNA and thus likely transcribing more *sls* mRNA. Sls protein is the titin homolog, a critical component of the sarcomeric Z-disks (*Loreau et al., 2023*), also located at the terminal Z-disk linking each force-probing myofibril to the tendon cells (*Green et al., 2018*). Similarly, in mammalian muscles, it was recently found that the terminal muscle fiber nuclei, which face the tendon cells, apply a specialized transcriptional program (*Kim et al., 2020*; *Esteves de Lima et al., 2021*). Interestingly, the FCA data of cross-tissue muscle cells did not reveal transcriptionally heterogeneous subclusters of nuclei, showing the benefit of more sensitive spatial techniques (*Li et al., 2022*).

To conclude, we have shown that medium throughput SRT on entire fly head and body samples is methodologically possible and leads to informative new discoveries that will spark interesting follow-up studies. All our data can be freely explored at https://spatialfly.aertslab.org. This proof-of-concept study opens avenues for potential full 3D slicing and imaging of an adult fly in its entirety at high resolution.

# Materials and methods

## Key resources table

| Reagent type (species) or resource | Designation | Source or reference | Identifiers | Additional information |
|---|---|---|---|---|
| strain, strain background (*Drosophila melanogaster*) | Luminy | *Loreau et al., 2023* | | |
| gene (*Drosophila melanogaster*) | *sls* | http://flybase.org/reports/FBgn0086906 | FBgn0086906 | |
| gene (*Drosophila melanogaster*) | *Act88F* | http://flybase.org/reports/FBgn0000047 | FBgn0000047 | |
| gene (*Drosophila melanogaster*) | *Act88F* | http://flybase.org/reports/FBgn0000047 | FBgn0000047 | |
| gene (*Drosophila melanogaster*) | *TpnC4* | http://flybase.org/reports/FBgn0033027 | FBgn0033027 | |
| gene (*Drosophila melanogaster*) | *Mhc* | http://flybase.org/reports/FBgn0264695 | FBgn0264695 | |
| chemical compound, drug | OCT | VWR | 00411243 | |
| chemical compound, drug | Gelatin | Merck | porcine skin, 300 g Bloom Type A | |
| software, algorithm | SCANPY | *Wolf et al., 2018*; https://github.com/scverse/scanpy | RRID:SCR_018139 Version: v.1.8.1 | |
| software, algorithm | Tangram | *Biancalani et al., 2021*; https://github.com/broadinstitute/Tangram | Version: v1.0.2 | |

*Continued on next page*

*Continued*

| Reagent type (species) or resource | Designation | Source or reference | Identifiers | Additional information |
|---|---|---|---|---|
| software, algorithm | SpaGE | *Abdelaal et al., 2020*; https://github.com/tabdelaal/SpaGE | | |
| software, algorithm | Spage2Vec | *Partel and Wählby, 2021*; https://github.com/wahlby-lab/spage2vec | Version: v2.0 | |
| software, algorithm | Cellprofiler | *Stirling et al., 2021*; https://cellprofiler.org/ | RRID:SCR_007358 Version: v4.0.7 | |
| software, algorithm | SpatialNF | https://github.com/aertslab/SpatialNF | | |

## Full fly sectioning for MC

To ensure full adult development, adult male Luminy flies were isolated and left for 3 d at 25 °C. After this step, the flies were put on ice for 15 min, and their wings were clipped. Flies were then transferred to a freezing mold alive, embedded in optimal cutting temperature compound (OCT compound, VWR), and frozen in liquid nitrogen. Flies were sectioned with a cryostat (Leica CM 3050 S, Leica Biosystems, Germany). 10 µm sections were transferred to coverslips coated with gelatin (porcine skin, 300 g Bloom Type A, Merck) and stored at –80 °C. Five body sections from one male were imaged with MC.

## Fly head sectioning for MC

Adult flies (male and females) were anesthetized on a fly pad using $CO_2$ gas, to cut off their heads with a scalpel. Heads were then placed on a pre-cooled metal surface on dry ice and covered with a drop of OCT. Frozen OCT blocks were stored at –80 °C until sectioning. Head sections of 10 µm thickness were produced with a Leica CM3050 S cryostat and placed on uncoated coverslips, which were stored at –80 °C. 13 head sections from 12 different flies were imaged with MC.

## Molecular Cartography (MC)

### Gene selection and probe design

Guided by the fly cell atlas, 150 genes were selected to be the most informative for spatial transcriptomics, 50 genes for the whole-body samples (*Supplementary file 1*) and 100 genes for the head samples (*Supplementary file 2*). These genes were selected based on different criteria including: marking specific cell populations, strong single-cell co-expression, marking uncharacterized cell populations, or showing broad expression.

The probes for the selected genes were designed using Resolve Biosciences' proprietary design algorithm. To speed up the process, the calculation of computationally expensive parts, especially the off-target searches, the selection of probe sequences was not performed randomly, but limited to sequences with high success rates. To filter highly repetitive regions, the abundance of k-mers was obtained from the background transcriptome using Jellyfish (*Marçais and Kingsford, 2011*). Every target sequence was scanned once for all k-mers and those regions with rare k-mers were preferred as seeds for full probe design. A probe candidate was generated by extending a seed sequence until a certain target stability was reached. A set of simple rules was applied to discard sequences that were found experimentally to cause problems. After these fast screens, every kept probe candidate was mapped to the background transcriptome using ThermonucleotideBLAST (*Gans and Wolinsky, 2008*), and probes with stable off-target hits were discarded. Specific probes were then scored based on the number of on-target matches (isoforms), which were weighted by their associated APPRIS level (*Rodriguez et al., 2018*), favoring principal isoforms over others. A bonus was added if the binding site was inside the protein-coding region. From the pool of accepted probes, the final set was composed by greedily picking the highest-scoring probes. Catalog numbers for the specific probes are available upon request at Resolve Biosciences.

### Molecular Cartography

Samples were then sent to Resolve Biosciences on dry ice for analysis. Upon arrival, tissue sections were thawed and rehydrated with isopropanol, followed by 1 min washes in 95% ethanol and then 70% ethanol at room temperature. The samples were used for Molecular Cartography (100-plex

combinatorial single-molecule fluorescence in situ hybridization) according to the manufacturer's instructions (protocol v.1.3; available for registered users), starting with the aspiration of ethanol and the addition of buffer DST1 followed by tissue priming and hybridization. Briefly, tissues were primed for 30 min at 37 °C followed by overnight hybridization of all probes specific for the target genes (see below for probe design details and target list). Samples were washed the next day to remove excess probes and fluorescently tagged in a two-step color development process. Regions of interest were imaged as described below and fluorescent signals were removed during decolorization. Color development, imaging, and decolorization were repeated for multiple cycles to build a unique combinatorial code for every target gene that was derived from raw images as described below.

## Imaging

Samples were imaged on a Zeiss Celldiscoverer 7, using the 50x Plan Apochromat water immersion objective with an NA of 1.2 and the 0.5x magnification changer, resulting in a 25x final magnification. Standard CD7 LED excitation light source, filters, and dichroic mirrors were used together with customized emission filters optimized for detecting specific signals. Excitation time per image was 1000 ms for each channel (4,6-diamidino-2-phenylindole (DAPI) was 20 ms). A z-stack was taken at each region with a distance per z-slice according to the Nyquist–Shannon sampling theorem. The custom CD7 CMOS camera (Zeiss Axiocam Mono 712) was used. For each region, a z-stack per fluorescent color (two colors) was imaged per imaging round. A total of eight imaging rounds were conducted for each position, resulting in 16 z-stacks per region. The completely automated imaging process per round (including water immersion generation and precise relocation of regions to image in all three dimensions) was realized by a custom Python script using the scripting API of the Zeiss ZEN software (open application development).

## Spot segmentation

The algorithms for spot segmentation were written in Java and are based on the ImageJ library functionalities. The iterative closest point algorithm is written in C++ based on the libpointmatcher library (https://github.com/ethz-asl/libpointmatcher, copy archived at *Pomerleau et al., 2024*).

## Preprocessing

As a first step, all images were corrected for background fluorescence. A target value for the allowed number of maxima was determined based on the area of the slice in µm² multiplied by the factor 0.5. This factor was empirically optimized. The brightest maxima per plane were determined, based upon an empirically optimized threshold. The number and location of the respective maxima were stored. This procedure was conducted for every image slice independently. Maxima that did not have a neighboring maximum in an adjacent slice (called a z group) were excluded. The resulting maxima list was further filtered in an iterative loop by adjusting the allowed thresholds for ($B_{abs}$ - $B_{back}$) and ($B_{peri}$ - $B_{back}$) to reach a feature target value ($B_{abs}$: absolute brightness; $B_{back}$: local background; $B_{peri}$: background of periphery within one pixel). These feature target values were based on the volume of the three-dimensional (3D) image. Only maxima still in a z-group of at least two after filtering were passing the filter step. Each z-group was counted as one hit. The members of the z-groups with the highest absolute brightness were used as features and written to a file. They resemble a 3D point cloud.

## Final signal segmentation and decoding

To align the raw data images from different imaging rounds, images had to be corrected. To do so, the extracted feature point clouds were used to find the transformation matrices. For this purpose, an iterative closest-point cloud algorithm was used to minimize the error between two-point clouds. The point clouds of each round were aligned to the point cloud of round one (the reference point cloud). The corresponding point clouds were stored for downstream processes. Based on the transformation matrices, the corresponding images were processed by a rigid transformation using trilinear interpolation. The aligned images were used to create a profile for each pixel consisting of 16 values (16 images from two color channels in eight imaging rounds). The pixel profiles were filtered for variance from zero normalized by the total brightness of all pixels in the profile. Matched pixel profiles with the highest score were assigned as an ID to the pixel. Pixels with neighbors having the same ID were

grouped. The pixel groups were filtered by group size, number of direct adjacent pixels in the group, and number of dimensions with a size of two pixels. The local 3D maxima of the groups were determined as potential final transcript locations. Maxima was filtered by the number of maxima in the raw data images where a maximum was expected. Remaining maxima were further evaluated by the fit to the corresponding code. The remaining maxima were written to the results file and considered to resemble transcripts of the corresponding gene. The ratio of signals matching to codes used in the experiment and signals matching to codes not used in the experiment were used as an estimate for specificity (false positives).

From Resolve Biosciences, the authors received the raw DAPI signal containing tiff image files, with gene localization count tables.

## Gene visualization

Genes were plotted using Python scripts. Marker sizes were scaled by gene density to increase the visibility of patterns.

## **Hybridization chain reaction on fly thoraces**

### Thorax preparation for HCR-FISH

Five day old Luminy males, raised at 25 °C, were put on ice for 15 min before clipping their wings, head, and abdomen. Thoraces were then transferred in PAXgene fixative (Resolve Biosciences) for 1 hr at room temperature, followed by 2 hr in stabilization buffer (Resolve Biosciences) at room temperature and overnight in 30% sucrose in 1 x PBS at 4 °C. The next day, thoraces were transferred to OCT (VWR) and immediately frozen in liquid nitrogen. Fly thoraces were sectioned with a cryostat (Leica CM 3050 S, Leica Biosystems, Germany). 16 µm sections were transferred to slides coated with gelatin (porcine skin, 300 g Bloom Type A, Merck).

### Probe design

Pairs of DNA 25-mer oligos were designed to hybridize on *Act88F*, *TpnC4*, *sls* and *Mhc* transcripts. Except for *TpnC4*, for which we designed 12 pairs of oligos, 20 pairs were designed for each mRNA, following the principles presented by *Choi et al., 2018*. Sequences are provided in *Supplementary file 4*.

### Hybridization chain reaction protocol

We followed the plated-cells protocol presented in *Choi et al., 2018* with slight modifications to apply it on fly sections.

Day 1: wash 2 x with PBS, add ice-cold 70% ethanol, and incubate overnight at –20 °C.

Day 2: aspirate ethanol, wash 2 x with 2 x SCC (Invitrogen), and incubate in 300 µL of '30% probe hybridization buffer' (Molecular Instruments) for 30 min at 37 °C. Incubate with 1.2 pmol of each probe mixture to 300 µL of '30% probe hybridization buffer' at 37 °C overnight.

Day 3: wash 4×5 min with 300 µL of '30% probe wash buffer' (Molecular Instruments) at 37 °C, and wash samples 2×5 min with 5 x SSCT (0.1% Tween 20 in 5 x SSC) at room temperature. Incubate samples in 300 µL of amplification buffer (Molecular Instruments) for 30 min at room temperature. Prepare 18 pmol of each fluorescently labeled hairpin (HCR Amplifiers B1-488 for *Act88F*, B3-546 for *TpnC4*, and B5-647 for *Sls* and Mhc, Molecular Instruments) by snap cooling (heat at 95 °C for 90 s and cool to room temperature in a dark drawer for 30 min). Prepare the hairpin solution by adding all snap-cooled hairpins to 300 µL of amplification buffer at room temperature. Incubate samples overnight (12–16 hr) in the dark at room temperature.

Day 4: wash 1 x with 300 µL of 5 x SSCT 5 min, stain with DAPI for 30 min, and wash for 5 min with 300 µL of 5 x SSCT at room temperature and mount in glycerol DABCO.

### Imaging of fly thoraces

Thoraces were imaged with a point scanning confocal microscope (Leica SP8) using a 20×0.75 NA glycerol immersion objective (Leica HC CS2 Plan Apo 20x0.75 NA Imm).

## Clustering analysis of mRNA species based on proximity

Based on the observation that groups of mRNA species appeared to cluster into separated regions of sections, we devised a simple method to automatically extract these regions based on the proximity of different mRNA. mRNA species that were identified as being close were then gathered into the same cluster.

In practice, we first computed the proximity of mRNA species by pair (see *Figure 2—figure supplement 3*). Each localization of two mRNA species was transformed into disks of fixed diameters, each disk being centered on a given mRNA localization; the diameter used here was 4 µm. To generate one surface for a species and avoid counting multiple times the same area, disks of a given mRNA species were merged if they were overlapping. We then computed their overlap surface from the surfaces obtained from two different mRNA species. The proximity of one mRNA species (mRNA$_1$) versus the other one (mRNA$_2$) was defined as the ratio between the overlap surface and the surface of the second mRNA species (mRNA$_2$):

$$\text{Proximity mRNA}_{1 \text{ in } 2} = \frac{\text{OverlapmRNA}_{1 \text{ and } 2}}{\text{Surface mRNA}_2}$$

Reciprocally,

$$\text{Proximity mRNA}_{2 \text{ in } 1} = \frac{\text{OverlapmRNA}_{1 \text{ and } 2}}{\text{Surface mRNA}_1}$$

The calculation of proximity then allowed us to define a distance between two mRNA species:

$$\text{Distance mRNA}_{1 \text{ and } 2} = 2 - \left(\text{Proximity mRNA}_{1 \text{ in } 2} + \text{Proximity mRNA}_{2 \text{ in } 1}\right)$$

mRNA species that show perfect overlap get a distance of 0 in this metric, whereas mRNA species that do not show any overlap would get a distance of 2.

Finally, this metric was used in a hierarchical clustering analysis using Ward's method. Clusters were then extracted from this analysis.

## 5 µm × 5 µm grid analysis

### Quantification

Samples were rasterized in a square grid of 36 by 36 pixels (1 pixel = 0.138 µm, 36 pixels = 4.968 µm). All counts within this area were summed up. This led to a square by gene matrix, with for every square the mean x and y spatial coordinates of the square's dimensions.

### SCANPY body

All body samples (5) were analyzed together in SCANPY (1.9.3) (*Wolf et al., 2018*). An increment of 1000 was added to both spatial x- and y-coordinates to arrange all samples together. Only squares with more than three counts were kept, leading to 132,642 squares with information for 50 genes. The data was subsequently normalized with 10,000 as size factor and log transformed. This matrix was then used as input for PCA, after which 40 components were retained by evaluating variance ratio plots. Harmony was then used to correct for batch effects between samples. 30 PCs were used to calculate neighbors, Leiden clustering (resolutions 0.2, 0.5, 1, and 2) and UMAP embeddings.

### SCANPY head

All head samples (13) were analyzed together in SCANPY (1.9.3) (*Wolf et al., 2018*). An increment of 1000 was added to both spatial x- and y-coordinates to arrange all samples together. Only squares with more than three counts were kept, leading to 83,064 squares with information for 99 genes (*ninaE* was discarded from most samples due to optical crowding). The data was subsequently normalized with 10,000 as size factor and log transformed. This matrix was then used as input for PCA, after which 30 components were retained by evaluating variance ratio plots. Harmony was then used to correct for batch effects between samples. 25 PCs were used to calculate neighbors, Leiden clustering (resolutions 0.2, 0.5, 1, and 2) and UMAP embeddings.

## Head OL vs CB vs PR differential expression

Leiden resolution 1 was used to create average gene expression profiles for clusters. Clusters with mean expression of para >0.05 were selected as neuronal clusters. These were subsequently manually assigned to either Photoreceptor (PR), Optic lobe (OL), or Central brain (CB) regions based on location. Next, the rank_genes_groups function from SCANPY was used to calculate differential genes for the regions based on a t-test.

## Gene set enrichment

In the body, we selected marker genes that were assigned to several categories. We then summed the expression of genes belonging to the same category to derive gene set signatures. The following gene sets were used: muscle (*Mhc*, *sls*, *CG32121*), neurons (*elav*, *Syt1*, *Sh*, *acj6*, *ey*, *VAChT*, *Gad1*, *VGlut*, *nAChRalpha7*), glia (*repo*, *alrm*), epithelia (pure: *grh*, *hth*, gut: *alphaTry*, *betaTry*), heart (*tin*, *Hand*), fat body (*AkhR*, *FASN2*), oenocyte (*FASN2*), male reproductive system (*Awh*, *eyg*, *svp*) and hemocytes (*Hml*). Spots in the grid were assigned to a category based on Z-scores. If z-normalized expression >1, the spot was assigned to the category. When conflicts arose, the following hierarchy was used: muscle > epithelia > glia > neurons > male reproductive system > fat body > oenocyte > hemocyte > heart.

## Lasso matching

We used lasso regression as implemented in sklearn (1.2.2). Averaged gene expression profiles of single-cell clusters were matched with expression profiles of the grid-based squares. The regression model was run using Lasso (alpha = 1, positive = True), forcing all coefficients to be positive, with all genes. Higher weights for the single-cell clusters correspond to a higher similarity between the cluster and the square. We only used weights >0.2 as confident matches. Spots were then assigned to the cluster based on the highest weight.

## Nuclei segmentation

Nuclei in DAPI images were segmented with Cellprofiler (version 4.0.7) (*Stirling et al., 2021*). Features of the DAPI images were enhanced (*EnhanceOrSuppressFeatures, Enhance, Speckles*) using a feature size of 100 pixels. Initial nuclei were detected using a size of 10–100 pixels, global Otsu thresholding with two classes, intensity for distinguishing clumped objects, *propagate* for drawing dividing lines. Initial nuclei were extended by 25 pixels. We then collected all transcripts inside a segmented cell to compute counts of each gene per segmented cell.

## Tangram

Tangram (version 1.0.2) (*Biancalani et al., 2021*) was used to project cell type labels from single-cell data. Here, we used the *cell* mapping mode to map single-cell data to each spatial slice separately. The mapping was computed on a NVIDIA A100-SXM4-80GB GPU. Prior to computing the mapping, segmented cells with less than three expressed genes and genes that were expressed in less than three cells were removed. Single cells and segmented cells were log-normalized in SCANPY (*Wolf et al., 2018*) with a target-sum of $10^4$. All genes that are shared between the spatial data and single-cell data after filtering were used for the mapping. Cell type labels for fly head samples were assigned separately for glia, optic lobe (OL), central brain (CB), and unknown clusters (UNK) by considering only the subset cell types that we grouped into the category. A cell type label was assigned to a segmented cell based on the 95%-quantile of the mapping scores for a cell type (unless stated otherwise) if this resulted in a unique label assignment. Cell type labels for the grid rasterized spatial data were assigned in the same manner as we did for the segmented cells. The only exception being, that we down-sampled the number of cells to a maximum of 5000 cells per cell type in the single-cell data to limit the amount of required video memory.

## Neighborhood embedding and SpaGE

Segmentation-free analysis of fly head and body SRT datasets have been carried out using spage2vec (*Partel and Wählby, 2021*). Briefly, spatial graphs of all mRNAs are first constructed for each different sample. Then, a graph convolution neural network trained in a self-supervised fashion projects each

mRNA into an embedding space based on its spatial neighborhood composition. Thus, mRNAs that share similar neighbors are mapped close together in the embedding space. Downstream clustering or visualization of mRNA embeddings unveil spatial gene expression patterns described by specific combinations of genes at subcellular resolution. Pseudo-cell counts have been generated for each mRNA by aggregating counts of $k$ neighboring mRNAs (i.e. k=100) in the embedding space. Finally, integration of reference scRNA-seq datasets with spatial pseudo-cell counts has been implemented by projecting both datasets into a common shared space using SpaGE (*Abdelaal et al., 2020*). Thereafter, cell type labels have been transferred from scRNA-seq cells to spatial pseudo-cells by kNN imputation (with k=100).

### mRNA localization in muscle - distance to nuclei

For every detected mRNA molecule, the distance to the closest nucleus was calculated. First, we calculated a mask to segment the indirect flight muscle using *Act88F* mRNA spots. Using opencv2, we performed a Gaussian blur (ksize = 5 5, sigma = 1), followed by two erosion steps (5×5 and 4×4) to remove sparse signals. Then, the spots were dilated thrice (50×50), followed by a final Gaussian blur (ksize = 5×5, sigma = 1). Only other mRNA species that overlap with this mask were kept for the distance analysis.

The nucleus segmentation was loaded as a black-white image. The nonzero function from opencv2 was used to find segmented pixels. Then for each mRNA spot, Euclidean distances were calculated to the segmented pixels, after which they were assigned to the closest pixel. To optimize the calculations, joblib's Parallel function was used. This was repeated over all body samples and results were combined. Only genes for which at least 100 mRNA spots were detected across samples and not localized at the edges (faulty segmentation along the muscle boundary) were kept: *Act88F*, *Mhc*, *TpnC4*, *sls*, *salm,* and *CG32121*.

## Acknowledgements

We thank VIB Tech Watch, Resolve Biosciences, and Jeroen Aerts for facilitating early access to the Molecular Cartography platform. Computing was performed on VSC (Vlaams Supercomputer Centrum). We thank Bruno C Vellutini for the help in guiding us through Hybridization Chain Reaction preparation and protocol. We thank Pavel Tomancak for hosting PM and providing support to perform HCR, and the MPI-CBG Light Microscopy Facility and Computer Services Facility for support. Funding: This work was in part funded by ERC AdG Genome2Cells (SA; 101054387); a FWO PhD Fellowship to (JJ; 1199518 N); a FWO Postdoctoral fellowship (NH; 1273822 N); GP was funded by Opening The Future. This work was supported by the Centre National de la Recherche Scientifique (CNRS, FS), Aix-Marseille University (PM), the European Research Council under the European Union's Horizon 2020 Programme (ERC-2019-SyG 856118 FS) and by funding from France 2030, the French Government program managed by the French National Research Agency (ANR-16-CONV-0001) and from Excellence Initiative of Aix-Marseille University - A*MIDEX (Turing Centre for Living Systems).

## Additional information

### Funding

| Funder | Grant reference number | Author |
|---|---|---|
| European Research Council | AdG Genome2Cells 101054387 | Stein Aerts |
| Horizon 2020 Framework Programme | ERC-2019-SyG 856118 | Frank Schnorrer |
| Research Foundation Flanders | PhD Fellowship 1199518N | Jasper Janssens |
| Research Foundation Flanders | Postdoctoral fellowship 1273822N | Nikolai Hecker |

| Funder | Grant reference number | Author |
|---|---|---|
| Opening The Future | | Nikolai Hecker<br>Gabriele Partel |
| Agence Nationale de la Recherche | ANR-16-CONV-0001 | Frank Schnorrer |
| Aix-Marseille Université | | Pierre Mangeol |
| Centre National de la Recherche Scientifique | | Frank Schnorrer |

The funders had no role in study design, data collection and interpretation, or the decision to submit the work for publication.

## Author contributions

Jasper Janssens, Conceptualization, Resources, Data curation, Software, Formal analysis, Validation, Investigation, Visualization, Methodology, Writing – original draft, Writing – review and editing; Pierre Mangeol, Resources, Data curation, Formal analysis, Validation, Investigation, Visualization, Methodology, Writing – original draft, Writing – review and editing; Nikolai Hecker, Gabriele Partel, Resources, Software, Formal analysis, Validation, Visualization, Methodology, Writing – review and editing; Katina I Spanier, Joy N Ismail, Data curation, Investigation, Methodology; Gert J Hulselmans, Resources, Software, Methodology; Stein Aerts, Conceptualization, Resources, Supervision, Funding acquisition, Visualization, Methodology, Writing – original draft, Project administration, Writing – review and editing; Frank Schnorrer, Conceptualization, Resources, Supervision, Funding acquisition, Visualization, Writing – original draft, Project administration, Writing – review and editing

## Author ORCIDs

Jasper Janssens (iD) https://orcid.org/0000-0001-9264-0782
Pierre Mangeol (iD) https://orcid.org/0000-0002-8305-7322
Nikolai Hecker (iD) https://orcid.org/0000-0003-1693-4257
Gabriele Partel (iD) https://orcid.org/0000-0002-4482-3119
Katina I Spanier (iD) https://orcid.org/0000-0002-1375-4157
Stein Aerts (iD) https://orcid.org/0000-0002-8006-0315
Frank Schnorrer (iD) https://orcid.org/0000-0002-9518-7263

Reviewer #1 (Public review): https://doi.org/10.7554/eLife.92618.3.sa1
Author response https://doi.org/10.7554/eLife.92618.3.sa2

# Additional files

## Supplementary files

MDAR checklist

Supplementary file 1. All 50 genes used for body MC experiments, including their expression levels and top expressing cell types inferred from the FCA.

Supplementary file 2. All 100 genes used for head MC experiments, including their expression levels and top expressing cell clusters inferred from the *Pech et al., 2024*.

Supplementary file 3. Detailed description of the 13 head section samples.

Supplementary file 4. All probe sequences used for the HCR-FISH experiments.

## Data availability

All data (raw DAPI .tiff files, mRNA localization tables) are available on https://spatialfly.aertslab.org/. *Supplementary files 1 and 2* list the genes used as probes. Probe design was performed by Resolve Biosciences. Gene panels are commercially available for use. HCR probes are provided in *Supplementary file 4*. Code for the website (https://spatialfly.aertslab.org/) is available on Github (https://github.com/aertslab/spatial_fly_website; *Janssens, 2024*). A pipeline containing scripts to perform grid, segmentation and neighborhood-embedding analysis can be found at https://github.com/aertslab/SpatialNF (copy archived at *Hecker, 2025*).

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
