## [Editor Report · eLife Assessment]

This **important** study presents a method to visualize the location of the cell types discovered through single-cell RNA sequencing. The data allowed the authors to build spatial tissue atlases of the fly head and body, and to identify the location of previously unknown cell types. The data are **convincing** and appropriate, and the authors validate the methodology in line with the current state-of-the-art.

---

## [Referee Report · Reviewer #1 (Public review)]

Summary:

In this manuscript, Janssens et al. addressed the challenge of mapping the location of transcriptionally unique cell types identified by single nuclei sequencing (snRNA-seq) data available through the Fly Cell Atlas. They identified 100 transcripts for head samples and 50 transcripts for fly body samples allowing identification of every unique cell type discovered through the Fly Cell Atlas. To map all of these cell types, the authors divided the fly body into head and body samples and used the Molecular Cartography (Resolve Biosciences) method to visualize these transcripts. This approach allowed them to build spatial tissue atlases of the fly head and body, to identify the location of previously unknown cell types and the subcellular localization of different transcripts. By combining snRNA-seq data from the Fly Cell Atlas with their spatially resolved transcriptomics (SRT) data, they demonstrated an automated cell type annotation strategy to identify uncharacterized clusters and infer their location in the fly body. This manuscript constitutes a proof-of-principle study to map the location of the cells identified by ever-growing single-cell transcriptomics datasets generated by others.

Strengths:

The authors used the Molecular Cartography (Resolve Biosciences) method to visualize 100 transcripts for head samples and 50 transcripts for fly body samples in high resolution. This method achieves high resolution by multiplexing a large number of transcript visualization steps and allows the authors to map the location of unique cell types identified by the Fly Cell Atlas.

Weaknesses:

Combining single-nuclei sequencing (snRNA-seq) data with spatially resolved transcriptomics (SRT) data is challenging, and the methods used by the authors in this study cannot reliably distinguish between cells, especially in brain regions where the processes of different neurons are clustered, such as neuropils. This means that a grid that the authors mark as a unique cell may actually be composed of processes from multiple cells.

Comments on revisions:

I believe the authors have improved the manuscript by addressing all the concerns and incorporating the suggestions raised by the reviewers. I have no further concerns or suggestions.

---

## [Author Response]

The following is the authors’ response to the original reviews.

**Public Reviews:**

**Reviewer #1 (Public Review):**
Summary:In this manuscript, Janssens et al. addressed the challenge of mapping the location of transcriptionally unique cell types identified by single nuclei sequencing (snRNA-seq) data available through the Fly Cell Atlas. They identified 100 transcripts for head samples and 50 transcripts for fly body samples allowing the identification of every unique cell type discovered through the Fly Cell Atlas. To map all of these cell types, the authors divided the fly body into head and body samples and used the Molecular Cartography (Resolve Biosciences) method to visualize these transcripts. This approach allowed them to build spatial tissue atlases of the fly head and body, to identify the location of previously unknown cell types and the subcellular localization of different transcripts. By combining snRNA-seq data from the Fly Cell Atlas with their spatially resolved transcriptomics (SRT) data, they demonstrated an automated cell type annotation strategy to identify unknown clusters and infer their location in the fly body. This manuscript constitutes a proof-of-principle study to map the location of the cells identified by ever-growing single-cell transcriptomic datasets generated by others.Strengths:The authors used the Molecular Cartography (Resolve Biosciences) method to visualize 100 transcripts for head samples and 50 transcripts for fly body samples in high resolution. This method achieves high resolution by multiplexing a large number of transcript visualization steps and allows the authors to map the location of unique cell types identified by the Fly Cell Atlas.

We thank this reviewer for appreciating the quality of our spatial data. We do not know what caused the technical problem (grayscale version of PDF) for this reviewer (the PDF figures are in color on the eLife website and on bioRxiv). We are surprised that the eLife discussion session did not resolve this issue.

Weaknesses:Combining single-nuclei sequencing (snRNA-seq) data with spatially resolved transcriptomics (SRT) data is challenging, and the methods used by the authors in this study cannot reliably distinguish between cells, especially in brain regions where the processes of different neurons are clustered, such as in neuropils. This means that a grid that the authors mark as a unique cell may actually be composed of processes from multiple cells.

The small size of an individual fly is one of the most challenging aspects of performing spatial transcriptomics. While the resolution of Molecular Cartography is rather high (< 200 nm), in the brain challenges remain as noted by the reviewer. *Drosophila* neuronal nuclei are notoriously small and cannot be easily resolved with the current imaging techniques. We agree that for a full atlas either expansion microscopy, 3D techniques or other super-resolution techniques will be required.

**Reviewer #2 (Public Review):**
Summary:The landmark publication of the "Fly Atlas" in 2022 provided a single cell/nuclear transcriptomic dataset from 15 individually dissected tissues, the entire head, and the body of male and female flies. These data led to the annotation of more than 250 cell types. While certainly a powerful and datarich approach, a significant step forward relies on mapping these data back to the organism in time and space. The goal of this manuscript is to map 150 transcripts defined by the Fly Atlas by FISH and in doing so, provide, for the first time, a spatial transcriptomic dataset of the adult fly. Using this approach (Molecular Cartography with Resolve Biosciences), the authors, furthermore, distinguish different RNA localizations within a cell type. In addition, they seek to use this approach to define previously unannotated clusters found in the Fly Atlas. As a resource for the community at large interested in the computational aspects of their pipeline, the authors compare the strengths and weaknesses of their approach to others currently being performed in the field.Strengths:(1) The authors use Resolve Biosciences and a novel bioinformatics approach to generate a FISHbased spatial transcriptomics map. To achieve this map, they selected 150 genes (50 body; 100 head) that were highly expressed in the single nuclear RNA sequencing dataset and were used in the 2022 paper to annotate specific cell types; moreover, the authors chose several highly expressed genes characteristic of unannotated cell types. Together, the approach and generated data are important next steps in translating the transcriptomic data to spatial data in the organism.

We thank the reviewer for this comment, as it reminded us that we need to be clearer in the text, about how we chose the genes to investigate. The statement that we selected “150 genes (50 body; 100 head) that were highly expressed in the single nuclear RNA sequencing dataset” is not correct. We have chosen genes with widely differing expression levels (log-scale range of 3.95 in body, 5.76 in head, we show this now in the new Figure 1 – figure fupplement 1B, D). Many of the chosen genes are also transcription factors. In fact, the here introduced method is more sensitive than the single cell atlas: the tinman positive cells were readily located (even non-heart cells were found to express tinman), whereas in the single cell FCA data tinman expression is often not detected in the cardiomyocytes (tinman is detected in 273 cells in the entire FCA (mean expression of 1.44 UMI in positive cells), and in 71 cells out of 273 cardiac cells (26%)).

(2) Working with Resolve, the authors developed a relatively high throughput approach to analyze the location of transcripts in *Drosophila* adults. This approach confirmed the identification of particular cell types suggested by the FlyAtlas as well as revealed interesting subcellular locations of the transcripts within the cell/tissue type. In addition, the authors used co-expression of different RNAs to unbiasedly identify "new cell types". This pipeline and data provide a roadmap for additional analyses of other time points, female flies, specific mutants, etc.(3) The authors show that their approach reveals interesting patterns of mRNA distribution (e.g alpha- and beta-Trypsin in apical and basal regions of gut enterocytes or striped patterns of different sarcomeric proteins in body muscle). These observations are novel and reveal unexpected patterns. Likewise, the authors use their more extensive head database to identify the location of cells in the brain. They report the resolution of 23 clusters suggested by the single-cell sequencing data, given their unsupervised clustering approach. This identification supports the use of spatial cell transcriptomics to characterize cell types (or cell states).(4) Lastly, the authors compare three different approaches --- their own described in this manuscript, Tangram, and SpaGE - which allow integration of single cell/nuclear RNA-seq data with spatial localization FISH. This was a very helpful section as the authors compared the advantages and disadvantages (including practical issues, like computational time).Weaknesses:(1) Experimental setup. It is not clear how many and, for some of the data, the sex of the flies that were analyzed. It appears that for the body data, only one male was analyzed. For the heads, methods say male and female heads, but nothing is annotated in the figures. As such, it remains unclear how robust these data are, given such a limited sample from one sex. As such, the claims of a spatial atlas of the entire fly body and its head ("a rosetta stone") are overstated. Also, the authors should clearly state in the main text and figure legends the sex, the age, how many flies, and how many replicates contributed to the data presented (not just the methods). What also adds to the confusion is the use of "n" in para 2 of the results. " ... we performed coronal sections at different depths in the head (n=13)..." 13 sections in total from 1 head or sections from 13 heads? Based on the body and what is shown in the figure, one assumes 13 sections from one head. Please clarify.

While we agree that sex differences present indeed an interesting opportunity to study with spatial transcriptomics, our goal was not to define male/female differences but rather to establish the technology to go into this detail if wanted in the future. In the revised version, we have provided an additional supplementary table with a more detailed description of the head sections (Table S3). We have added the number of animals (12 for the head sections, mixed sex; and 1 male for the body sections) to the main text. We would like to point out that we verified the specificity of our MC method on all the 5 body sections (Figure 2A, TpnC4 & Act88F and text) and not only on one. Furthermore, we also would like to state that the idea of “a Rosetta stone” was mentioned as a future prospect that clearly goes beyond our presented work. We have rewritten the discussion to make this clearer and to any avoid overstatements.

(2) Probes selected: Information from the methods section should be put into the main text so that it is clear what and why the gene lists were selected. The current main text is confusing. If the authors want others to use their approach, then some testing or, at the very least, some discussion of lower expressed genes should be added. How useful will this approach be if only highly expressed genes can be resolved? In addition, while it is understood that the company has a propriety design algorithm for the probes, the authors should comment on whether the probes for individual genes detect all isoforms or subsets (exons and introns?), given the high level of splicing in tissues such as muscle.

As stated above, while there is a slight bias to higher expressed genes (as expected for marker genes), we have also used low expressed genes like salm, CG32121, tinman (body) or sens (head). This is now shown in new Figure 1 – figure Supplement 1B, D. This shows that our method is more sensitive than single-cell data, as all cardiomyocytes can be identified by tinman expression and not only some are positive, as is the case in the FCA data. In fact, the method cannot resolve too highly expressed genes due to optical crowding of the signal leading to a worse quantification. For this reason, ninaE was removed from the analysis (as mentioned in Spatial transcriptomics allows the localization of cell types in the head and brain and in Methods).

As mentioned in the Methods, the probes are designed on gene level targeting all isoforms, but favoring principal isoforms (weighted by APPRIS level). The high level of splicing is indeed interesting and we expect that in the future spatial transcriptomics can help to generate more insight into this by designing isoform-specific probes.

(3) Imaging: it isn't clear from the text whether the repeated rounds of imaging impacted data collection. In many of what appear to be "stitched" images, there are gradients of signal (eg, figure 2F); please comment. Also, since this a new technique, could a before and after comparison of the original images and the segmented images be shown in the supplemental data so that the reader can better appreciate how the authors assessed/chose/thresholded their data? More discussion of the accuracy of spot detection would be helpful.

High-resolution imaging (pixel size = 138 nm) of a large field of view (>1mm) for spatial transcriptomics uses a stitching method to combine several individual images to reconstruct a large field of view. This does not generate signal gradients, apart from lower signal at the extreme edges of each of the individual images, as seen in our images, too. The spot detection algorithm was written and used by Resolve Biosciences and benchmarked for human (Hela) and mouse (NIH-3T3) cell lines in Groiss et al. 2021 (Highly resolved spatial transcriptomics for detection of rare events in cells, bioR xiv). The specificity of the decoded probes was found to lie between 99.45 and 99.9% here, matching the results we found for specific detection of TpnC4 and Act88F (99.4 and 99.8%).

(4) The authors comment on how many RNAs they detected (first paragraph of results). How do these numbers compare to the total mRNA present as detected by single-cell or single-nuclear sequencing?

We can compare the numbers, but the different methodologies make the interpretation of such a comparison difficult. FCA used single nucleus sequencing, so only nuclear pre-mRNAs are detected. The total amount of counts per single cell sample strongly depends on how many cells were sequenced in an experiment. MC detects all mRNAs present in the section. Here, the size of the sample and hence the size or the number of cells analyzed determines how many mRNAs are detected. In Author response image 1, we have compared our MC results versus FCA data, comparing the genes investigated here in MC per section vs per sequencing experiment. Numbers for MC are slightly lower for the brain (not all cell types are on all sections) and much higher for the larger body samples. However, we feel a direct comparison is questionable, so we prefer to not include this figure in our manuscript.

**Author response image 1. sa2fig1:** Barplots showing total number of mRNA molecules detected in Molecular Cartography (MC, Resolve, spatial spots) and in snRNA-seq data from the Fly Cell Atlas (10x Genomics, UMIs). Individual black dots show individual experiments, counts are only shown for the chosen gene panel for each sample. Bar shows the mean, with error bars representing the standard error.

(5) Using this higher throughput method of spatial transcriptomics, the authors discern different cell types and different localization patterns within a tissue/cell type.a. The authors should comment on the resolution provided by this approach, in terms of the detection of populations of mRNAs detected by low throughput methods, for example, in glia, motor neuron axons, and trachea that populate muscle tissue. Are these found in the images? Please show.

We did not add any markers for trachea in our gene panel, but we do detect sparse spots of repo (glia) and elav/VGlut in the muscle tissues (Gad1/VAChT are hardly detected in the muscle tissue). This is consistent with the glutamatergic nature of motor neurons in *Drosophila* as described previously (Schuster CM (2006), Glutamatergic synapses of *Drosophila* neuromuscular junctions: a high-resolution model for the analysis of experience-dependent potentiation. Cell Tissue Res 326: 287–299.). We have present these new data in new Figure 2 – figure supplement 1.

b.The authors show interesting localization patterns in muscle tissue for different sarcomere proteincoding mRNAs, including enrichment of sls in muscle nuclei located near the muscle-tendon attachment sites. As this high throughput approach is newly being applied to the adult fly, it would increase confidence in these data, if the authors would confirm these data using a low throughput FISH technique. For example, do the authors detect such alternating "stripes" (Act 88F, TpnC4, and Mhc) or enriched localization (sls) using FISH that doesn't rely on the repeated colorization, imaging, decolorization of the probes?

We thank the reviewer for the interest in the localization patterns in muscle tissue. We show that Act88F, TpnC4 are not detected outside of flight muscle cells (99.4% and 99.8% of the single molecular signal in flight muscles only), giving us confidence in the specificity of the MC method. Following the suggestion of the reviewer, we have adapted an HCR-FISH method to *Drosophila* adult body sections for the revised version of the manuscript. Using this method, we were able to confirm the higher expression/localization of sls transcripts to and around the adult flight muscle nuclei, with an enrichment in nuclei close to the muscle-tendon attachment sites (new Figure 4D-F and new Figure 4 – figure supplement 1). We have also been able to confirm some complementarity in the localization patterns of Act88F and TpnC4 in longitudinal stripes in adult flight muscles, however for Mhc we could not confirm this pattern with HCR-FISH (new Figure 5C-F and new Figure 5 – figure supplement 1). While we could confirm most of the pattern seen, we do not know the exact reason for the slight discrepancies. Thus, we now recommend that insights found with SRT should be confirmed with more classical FISH methods.

(6) The authors developed an unbiased method to identify "new cell types" which relies on coexpression of different transcripts. Are these new cell types or a cell state? While expression is a helpful first step, without any functional data, the significance of what the authors found is diminished. The authors need to soften their statements.

The term “new cell types” only appeared in the old title. We agree that with the current spatial map we cannot be sure to have found “new cell types”, instead we show where unannotated/uncharacterized clusters from the scRNA-seq atlas are located, based on their gene expression. Therefore, we have updated the title in the revised version (Spatial transcriptomics in the adult *Drosophila* brain and body) and thank the reviewer for this valuable suggestion.

Appraisal:The authors' goal is to map single cell/nuclear RNAseq data described in the 2022 Fly Atlas paper spatially within an organism to achieve a spatial transcriptomic map of the adult fly; no doubt, this is a critical next step in our use of 'omics approaches. While this manuscript does the hard work of trying to take this next step, including developing and testing a new pipeline for high throughput FISH and its analysis, it falls short, in its present form, in achieving this goal. The authors discuss creating a robust spatial map, based on one male fly. Moreover, they do not reveal principles of mRNA localization, as stated in the abstract; they show us patterns, but nothing about the logic or function of these patterns. This same criticism can be said of the identification of "new cell types, just based on RNA colocalization. In both cases (mRNA subcellular localization or cell type identification), further data in the form of validation with traditional low throughput FISH and genetic manipulations to assess the relation to cell function are required for the authors to make such claims.

We have indeed used one male fly for the adult male body data. This is mainly due to the cost of the sample processing. We used 12 individuals for the head samples (from 1 individual we acquired 2 sections, a total of 13 sections). We show that the body samples show a high correlation with each other, while the head samples cover multiple depths of the head. Still, even in the head, we find that sections at similar depths show a high similarity to each other in terms of gene-gene coexpression and expression patterns. Although obtaining sections from more animals would be valuable, we do not believe it to be necessary for our current goals. Additional replicates beyond the ones we already provide would require significant amounts of extra time and budget, while they would very likely produce similar results as we already show. Following the reviewer’s suggestion, we have tested several genes with HCR-FISH and could readily confirm the localization pattern of sls mRNA close to the terminal nuclei of the flight muscles. This pattern is likely due to a higher expression of sls in these nuclei, as a large amount of sls mRNA signal is detected within the nuclei (Figure 4). A detailed dissection of the mechanism that establishes this pattern is beyond the scope of this manuscript, which is the first one on applying spatial transcriptomics to adult *Drosophila*.

The usage of the term “new cell types” was indeed ambiguous and we removed this from the revised version. We now clarified that we map the spatial location of unannotated clusters in the brain. This may or may not include uncharacterized cell types. We now further specify that we have only inferred the location of the nuclei; thus, neuronal function or the location of their axonal processes are still unknown. As such, our data provides a starting point to identify uncharacterized cell types, since their marker genes and nuclear location are now determined. The next step to identify “new cell types” would indeed be to acquire genetic access to these cell types and characterize them in more detail. This is beyond the scope of this manuscript, and therefore we have toned down the title in the revised version and thank the reviewer for this valuable suggestion.

Discussion of likely impact:If revised, these data, and importantly the approach, would impact those working on *Drosophila* adults as well as those working in other model systems where single cell/nuclear sequencing is being translated to the spatial localization within the organism. The subcellular localization data - for example, the size of transcripts and how that relates to localization or the patterns of sarcomeric protein localization in muscle - are intriguing, and would likely impact our thinking on RNA localization, transport, etc if confirmed. Lastly, the authors compare their computational approaches to those available in the field; this is valuable as this is a rapidly evolving field and such considerations are critical for those wishing to use this type of approach.

We thank this reviewer for appreciating the impact of our findings and approach to the *Drosophila* field and beyond. We here provide the groundwork for a full *Drosophila* adult spatial atlas, similar to how early scRNA-seq datasets provided a framework for the Fly Cell Atlas. In the manuscript we provide both experimental information on how to successfully perform spatial transcriptomics (treating slides for optimal attachment) and the data serves as a benchmark for future experiments to improve upon (similar to how early Drop-seq datasets were compared to later 10x datasets in single-cell transcriptomics). In addition, it also provides proof of principle methods on how to integrate the FCA data with these spatial data and it identifies localized mRNA species in large adult muscle cells, showing the complementarity of spatial techniques with single-cell RNA-seq. For a small number of genes, we have confirmed the mRNA patterns using HCR-FISH in the revised version of this manuscript. To conclude, this is the first spatial adult *Drosophila* transcriptomics paper, locating 150 mRNA species with easy data access in our user portal (https://spatialfly.aertslab.org/).

**Recommendations for the authors:**

**Reviewer #1 (Recommendations For The Authors):**
(1) All figures in the manuscript were in grayscale, which made it difficult to interpret the results because the data could only be interpreted by distinguishing different colors to visualize different transcripts. This is likely a technical problem. The manuscript must contain colored images.

We apologize to the reviewer for this technical issue. The manuscript was uploaded in color to bioRxiv and to eLife. We therefore do not understand to reason for this problem. We are surprised that this issue was not resolved in the reviewers’ discussion since color is obviously essential to appreciate the beauty of this manuscript.

(2) In Figure 2a, the authors comment on the subcellular localization of trypsin isoforms, but the figure does not indicate the cell borders or the apical and basal regions of the cell. These must be indicated in the figure to help readers understand the results.

We thank the reviewer for pointing this out; we have now indicated the outlines of the single-cell layer epithelium on the figure. While we have no marker for cell borders, we have a nuclear marker showing that it is a single cell layer. We hope this allows the reader to appreciate the subcellular localization of the trypsin isoforms.

(3) All figures (including the data on the authors' website) contain background staining, which I assume is labeling nuclei. This is not indicated in the manuscript, and should be clarified.

We again thank the reviewer for pointing this out; the background staining indeed labels nuclei (using DAPI). We have indicated this better in the revised version.

(4) In Figure 5c, the authors claim that neuronal and muscular genes are grouped into the same cluster, but they do not indicate which transcripts are neuronal and which ones are muscular. This must be indicated in the figure.

We thank the reviewer for this comment. Indeed, there was only one gene, acj6, present in the muscle cluster. So, we decided to delete this statement in the revised version.

(5) The authors utilized and compared three different approaches to integrate single nuclei sequencing data from the Fly Cell Atlas to their spatially resolved transcriptomics (SRT) data. I was wondering if it is possible to generate a virtual expression explorer using this integrated data, similar to the dataset published in the 2017 Science article by Karaiskos et al., where they combined publicly available in situ hybridization data of fly embryos and their single-cell sequencing data. This virtual expression explorer would be useful to visualize the expression pattern of transcripts that the authors of this paper did not use for their SRT.

We thank the reviewer for this interesting comment. Using Tangram, we indeed infer gene expression for all genes from the Fly Cell Atlas. To make this visible we have created a Scope session (https://scope.aertslab.org/#/Spatial_Fly/*/welcome), with which users can browse inferred gene expression levels (note that this is on a segmented cell level). We do notice that the inferred gene expression levels contain many false positives and should therefore be used with caution. The spatial data themselves can be browsed through the spatial portal at https://spatialfly.aertslab.org/ .

**Reviewer #2 (Recommendations For The Authors):**
Suggestions for improved or additional experiments, data, or analyses:The authors have used a new high throughput approach to examine the location of 150 RNAs in adult *Drosophila* heads or one body. It is unclear whether the fixation/repeated imaging etc is accurately reflecting the patterns of expression in vivo. The authors should confirm these data using low throughput established techniques for the RNA patterns in muscle for example.The authors should clarify their experimental approaches and include additional samples if they indeed want to establish the rosetta stone of fly adults. These data are from only a male fly (and as such is not a complete analysis of the adult fly). To be a map of the adult fly, data from both sexes need to be included.Unless functional data that complement the descriptive data shown here are included, the authors have to soften their conclusions. For example, while spatial transcriptomics has mapped RNA expression to a location, without some functional data, it is difficult to conclude that these are indeed "new cell types". Same with the RNA localization principles.Recommendations for improving the writing and presentation:(1) The manuscript should be heavily revised: in many places, important details are left out or should be moved from the methods to the main text. In addition, the authors often overstate their findings throughout the manuscript. As an example, it appears that the data presented is only from 1 fly, so this doesn't increase the reader's confidence in the data or the applicability of the approach. Also, it isn't clear how many flies were analyzed for the heads (one male fly too?) nor what variability is present from fly to fly. For the approach and data to be used by others, this is important to know.

We moved some text from the methods section to the main text to be clearer. We now also state how many animals were used for the MC method. While the data for the body has been generated from 1 male only, the data for the head was generated from 12 flies; for both cases, similar slices show very similar gene expression patterns. Furthermore, in the body we used widely known and published marker genes that all showed expected expression patterns, indicating robustness. We agree that this is not a full spatial atlas of the fly, this was also not our goal and we have removed such general statements from the revised version: we aimed to generate a spatial transcriptomics dataset, covering the entire fly (head and body) as a proof-of-principle, tackling data generation and analysis, and highlighting challenges in both.

(2) The grammar and word choice throughout are challenging often making the text difficult to follow. This reads like an early draft of the paper.

We apologize to the reviewer for any difficulties. We have revised the text and hope it is now easier to read, while still being accurate on the technical details of the various methods used in our manuscript.

Minor corrections to the text and figures.See the weaknesses mentioned above. Also:Figure S1 is unreadable.

There is no simple way to describe the expression values of 100 genes in 100 cell types on a single page. The resolution of the PDF is high enough that after zooming in, all the information can be read easily.

Figure S2, in a, please include the axes so that the reader can better understand the sections shown.In b, it is unclear what the pink boxes mean. In c, the labels are barely legible.

In Figure 1 – figure supplement 2 (head sections), we have ordered the head sections from anterior to posterior. The boxes in (B) represent boxplots. We have updated this plot for clarity to better display the number of mRNA molecules detected for each gene. We have increased the font size in (C).

Figure S3, in a, please include axes. In b, the meaning of the pink box

In Figure 1 – figure supplement 3 (the body sections) we have added the anterior to posterior and dorso-ventral axis, and ordered the sections that stem from the same animal. The boxes in (B) represent boxplots. We have updated this plot for clarity to better display the number of mRNA molecules detected for each gene. We have added an explanation to the figure legend.

Figure S4, the text in the axes of the heatmap should have a darker typeface

We have changed it to black, thanks.

Figure S5c, are the colors in the dendrogram supposed to match the spatial location on the right?The purple of the muscles is barely visible.

Yes, they do match. Colors were modified in the revised version for better visibility.